# Toxoplasma TgATG9 is critical for autophagy and long-term persistence in tissue cysts

David Smith[1]*, Geetha Kannan[1], Isabelle Coppens[2], Fengrong Wang[1], Hoa Mai Nguyen[3], Aude Cerutti[3], Einar B Olafsson[1], Patrick A Rimple[1], Tracey L Schultz[1], Nayanna M Mercado Soto[1], Manlio Di Cristina[1,4], Sébastien Besteiro[3], Vern B Carruthers[1]*

[1]Department of Microbiology and Immunology, University of Michigan Medical School, Ann Arbor, United States; [2]Department of Molecular Microbiology and Immunology, Johns Hopkins University Bloomberg School of Public Health, Baltimore, United States; [3]Laboratory of PathogenHost Interactions, UMR 5235, CNRS, Université de Montpellier, Montpellier, France; [4]Department of Chemistry, Biology and Biotechnology, Università degli Studi di Perugia, Perugia, Italy

**Abstract** Many of the world's warm-blooded species are chronically infected with *Toxoplasma gondii* tissue cysts, including an estimated one-third of the global human population. The cellular processes that permit long-term persistence within the cyst are largely unknown for *T. gondii* and related coccidian parasites that impact human and animal health. Herein, we show that genetic ablation of *TgATG9* substantially reduces canonical autophagy and compromises bradyzoite viability. Transmission electron microscopy revealed numerous structural abnormalities occurring in Δ*atg9* bradyzoites. Intriguingly, abnormal mitochondrial networks were observed in TgATG9-deficient bradyzoites, some of which contained numerous different cytoplasmic components and organelles. Δ*atg9* bradyzoite fitness was drastically compromised in vitro and in mice, with very few brain cysts identified in mice 5 weeks post-infection. Taken together, our data suggests that TgATG9, and by extension autophagy, is critical for cellular homeostasis in bradyzoites and is necessary for long-term persistence within the cyst of this coccidian parasite.

*For correspondence:
d.smith@moredun.ac.uk (DS);
vcarruth@umich.edu (VBC)

**Competing interests:** The authors declare that no competing interests exist.

## Introduction

The subclass Coccidiasina (hereafter coccidia) includes several notable parasites that are important to public and animal health (*Smith, 1981*). A unifying feature of coccidia is their ability to form persistent cysts. All coccidian parasites generate environmentally resilient oocysts that persist in soil or water for fecal-oral transmission (*Belli et al., 2006*; *Frenkel and Smith, 2003*). Also, some coccidia (those of the family Sarcocystidae, meaning 'flesh cyst') including *Toxoplasma gondii* form long-lived tissue cysts that confer transmission via carnivorism (*Frenkel and Smith, 2003*). Coccidia are further classified within the phylum Apicomplexa along with haemosporidians including Plasmodium spp. and Babesia spp., the agents of malaria and babesiosis, respectively (*Mathur et al., 2019*). Because *T. gondii* is genetically tractable, can be grown continuously in culture, and can be injected into mice for a reliable model of the disease, this parasite has become an attractive model system for other related parasites. *T. gondii* is also recognized as an important public health pathogen, acquired through the ingestion of tissue cysts in contaminated meat, or water and food contaminated with oocysts (*Ortega, 2019*). Infection with this parasite can have serious negative health consequences (*Montoya and Liesenfeld, 2004*). If acquired early during pregnancy, the parasite can traverse the placenta and infect the developing fetus, which can result in miscarriage and neurological

pathologies (*Bresciani and da Costa, 2018*). Infection can be fatal in immunocompromised individuals, for example, AIDS patients, organ transplant recipients, or patients undergoing chemotherapy (*Montoya and Liesenfeld, 2004*). There have also been reports of emerging *T. gondii* strains causing complications and death in otherwise healthy individuals (*Bresciani and da Costa, 2018*; *Carme et al., 2009*).

Within an intermediate host, sarcocystidian parasites exist in both an acute stage (tachyzoite) and a chronic stage tissue cyst (bradyzoite). During the tachyzoite phase, *T. gondii* parasites replicate rapidly within a parasitophorous vacuole (PV). Throughout the acute stage of infection, nutrients are trafficked from the host cytosol across the PV membrane (*Dou et al., 2014*; *McGovern et al., 2018*). The removal of amino acids from culture media in vitro triggers tachyzoite to bradyzoite differentiation (*Fox et al., 2004*; *Ferreira da Silva et al., 2008*). In an in vivo setting within an immunocompetent intermediate host, an effective host response involves CD8+ T cell recognition of host cells infected with *T. gondii* and production of interferon gamma (*Fisch et al., 2019*; *Suzuki et al., 2011*). In turn, this results in the restriction of amino acids, including arginine and tryptophan, in cells harboring the parasite (*Pfefferkorn et al., 1986*; *Coppens, 2014*; *Lüder and Rahman, 2017*). As parasites continue to draw such resources, the infected host cell might become an increasingly nutrient-limited environment. Accordingly, it has been suggested that constrained access to essential nutrients is a major factor driving tachyzoite to bradyzoite differentiation in vivo (*Lüder and Rahman, 2017*). However, this also raises the question as to how *T. gondii* tissue cysts containing bradyzoites are able to persist within the host with potentially limited access to host material for sustenance. Although recent work reported that *T. gondii* bradyzoites can ingest and digest protein from the cytosol of infected cells in vitro (*Kannan et al., 2021*), the extent to which this process helps to sustain bradyzoites remains unknown.

Macroautophagy (hereafter autophagy, for 'self-eating') is an intracellular catabolic process that facilitates the encapsulation, trafficking, and degradation of endogenous proteins and organelles (*Yin et al., 2016*; *Suzuki et al., 2017*). This process begins with the activation of an initiation complex comprised of several proteins including the protein kinase ATG1. ATG1 activity catalyzes several downstream events including the assembly of a complex comprising the lipid transfer protein ATG2, the PROPPIN family protein ATG18, and the only integral membrane protein in the pathway ATG9. The ATG2/18/9 complex facilitates the development of a double-membraned phagophore that elongates while capturing cytoplasmic material. Recent studies have reported that ATG2 is a lipid transfer protein and ATG9 functions as a scramblase that flips phospholipids from the outer to inner leaflet of the developing phagophore (*Maeda et al., 2020*; *Orii et al., 2021*; *Matoba et al., 2020*; *Matoba and Noda, 2020*; *Valverde et al., 2019*; *Osawa et al., 2019*). During the process of elongation, the small ubiquitin-like protein ATG8 accumulates on the developing phagophore via lipid conjugation to phosphatidylethanolamine. The phagophore closes to generate a double-membrane vesicle, the autophagosome. Fusion of the autophagosome with a lytic compartment such as the lysosome (vertebrates) or the vacuole (yeast and apicomplexan parasites) creates an autolysosome within which cytoplasmic material is degraded by cathepsin proteases and other hydrolytic enzymes. The entire process of development, closure, and fusion occurs quite rapidly, often within 3–10 min (*Xie et al., 2008*; *Mesquita et al., 2017*).

The turnover of autophagic material via the digestive organelle can serve a number of purposes that all contribute toward cellular homeostasis and promote cell survival. These include recycling nutrients as part of a starvation response, removal of damaged material from the cytoplasm, turnover of proteins and organelles during developmental changes, and the removal of pathogens (*Yin et al., 2016*). The autophagy machinery is widely conserved among eukaryotes. However, whether or not all Apicomplexa are able to perform canonical autophagy is still a matter of debate, even if there are clues *T. gondii* has a functional pathway (*Besteiro, 2017*). Interestingly, several Plasmodium and Toxoplasma autophagy-related proteins are also involved in a non-canonical function related to the segregation of the apicoplast (*Nguyen et al., 2018*; *Lévêque et al., 2015*; *Walczak et al., 2018*; *Bansal et al., 2017*), a non-photosynthetic plastid shared by most members of the phylum.

It has previously been shown that while *T. gondii* cathepsin L (TgCPL) is dispensable in tachyzoites (*Dou et al., 2014*; *Smith et al., 2020*), inhibition or genetic ablation of this protease in bradyzoites results in parasite death (*Smith et al., 2020*; *Di Cristina et al., 2017*). TgCPL is a cysteine protease that resides within a plant-like vacuole/vacuolar compartment (hereafter VAC) in *T. gondii*, wherein it

is a major enzyme required for the degradation of proteinaceous material (*Dou et al., 2014*; *McGovern et al., 2018*; *Smith et al., 2020*; *Di Cristina et al., 2017*). Interestingly, we previously demonstrated the accumulation of autophagic material and organellar remnants within the VAC in parasites lacking active TgCPL (*Di Cristina et al., 2017*). Collectively, these findings suggested the accumulation of undigested autophagic material within the VAC in bradyzoites was somehow contributing to parasite death, although precisely why parasites were dying remained unknown (*Di Cristina et al., 2017*).

In the present study, we set out to explain why TgCPL-deficient bradyzoites fail to survive and to determine the extent to which autophagy itself directly contributes to chronic persistence of *T. gondii* as a model sarcocystidian and coccidian parasite.

## Results

### Conservation of canonical autophagic proteins in coccidian parasites

Previous analyses of the autophagy machinery in *T. gondii* have shown a reasonably high degree of evolutionary conservation compared with other apicomplexan parasites (*Besteiro, 2017*; *Lévêque and Besteiro, 2016*). Here, we sought to determine the relative conservation of *T. gondii* autophagy genes among related apicomplexans, particularly cyst-forming parasites, as well as related but non-parasitic aveolates. Expectedly, we found that autophagy-related genes in the closely related sarcocystidian parasites *Hammondia hammondi* and *Neospora caninum* are most conserved with those of *T. gondii* (*Figure 1A*). Intriguingly, while some genes associated with canonical autophagy (e.g., ATG2 and ATG9) are also conserved in several coccidian parasites, they were not identified in the non-coccidian apicomplexan parasites *Cryptosporidium parvum*, *Plasmodium falciparum*, and Babesia spp. Furthermore, autophagy-related genes tended to be more conserved between *T. gondii* and the non-parasitic aveolates *Chromera velia* and *Vitrella brassicaformis* (the closest known autotrophic organisms to the Apicomplexa) than between *T. gondii* and other non-coccidian Apicomplexa. However, matches to the autophagy-related *T. gondii* genes that are not only involved in canonical autophagy but also in maintenance of the apicoplast (which is found in all apicomplexans except Cryptosporidium spp.), namely ATG3, ATG7, ATG8, Prop2, VPS15, and VPS34, were found across all the protists considered in this analysis (except in Babesia spp., which only had ATG7, ATG8, and VPS34). Taken together, this information indicates a potential role for canonical autophagy in sarcocystidia and possibly some other coccidia versus a more universal non-canonical function for autophagy-related proteins in maintenance of the apicoplast.

To gain a sense of which components of the autophagy pathway in *T. gondii* play critical roles for the parasite, we mapped such constituents to a working model of autophagy and color-coded them according to their phenotype scores from a recent genome-wide CRISPR-Cas9 screen (*Sidik et al., 2016*; *Figure 1B*). The phenotype score reflects the relative fitness cost associated with loss of each gene during in vitro replication of tachyzoites. A strongly negative phenotype score suggests a gene is likely to be broadly indispensable, whereas a neutral or positive phenotype score suggests dispensability, at least during acute stage replication in culture. Interestingly, proteins that have been implicated in segregation of the apicoplast, including TgATG8 and its lipid conjugation machinery (e.g., TgATG3, TgATG5, TgATG7, and TgATG12), together with most proteins in the VPS34/PI3P kinase complex, have strongly negative phenotype scores. Conversely, proteins that are theoretically (TgATG1 and TgATG2) or empirically (TgProp1 [*Nguyen et al., 2018*] and TgATG9 [*Lévêque et al., 2015*]) linked exclusively to canonical autophagy have neutral or positive phenotype scores (*Figure 1B*). This is consistent with canonical autophagy playing a non-essential role in cultured tachyzoites. Nevertheless, that proteins involved exclusively in canonical autophagy have been retained in cyst-forming parasites suggests that their biological significance could manifest in a different stage of the infection.

### TgATG8 is required for autophagy and viability of *T. gondii* bradyzoites

We previously demonstrated that autophagic material and lipidated TgATG8 accumulate in TgCPL-deficient bradyzoites due to lower protein turnover in the VAC (*Di Cristina et al., 2017*). Although this suggested that bradyzoites undergo autophagy, the evidence was indirect. As an initial strategy to directly test for autophagy in bradyzoites, we targeted TgATG8, which is necessary for apicoplast

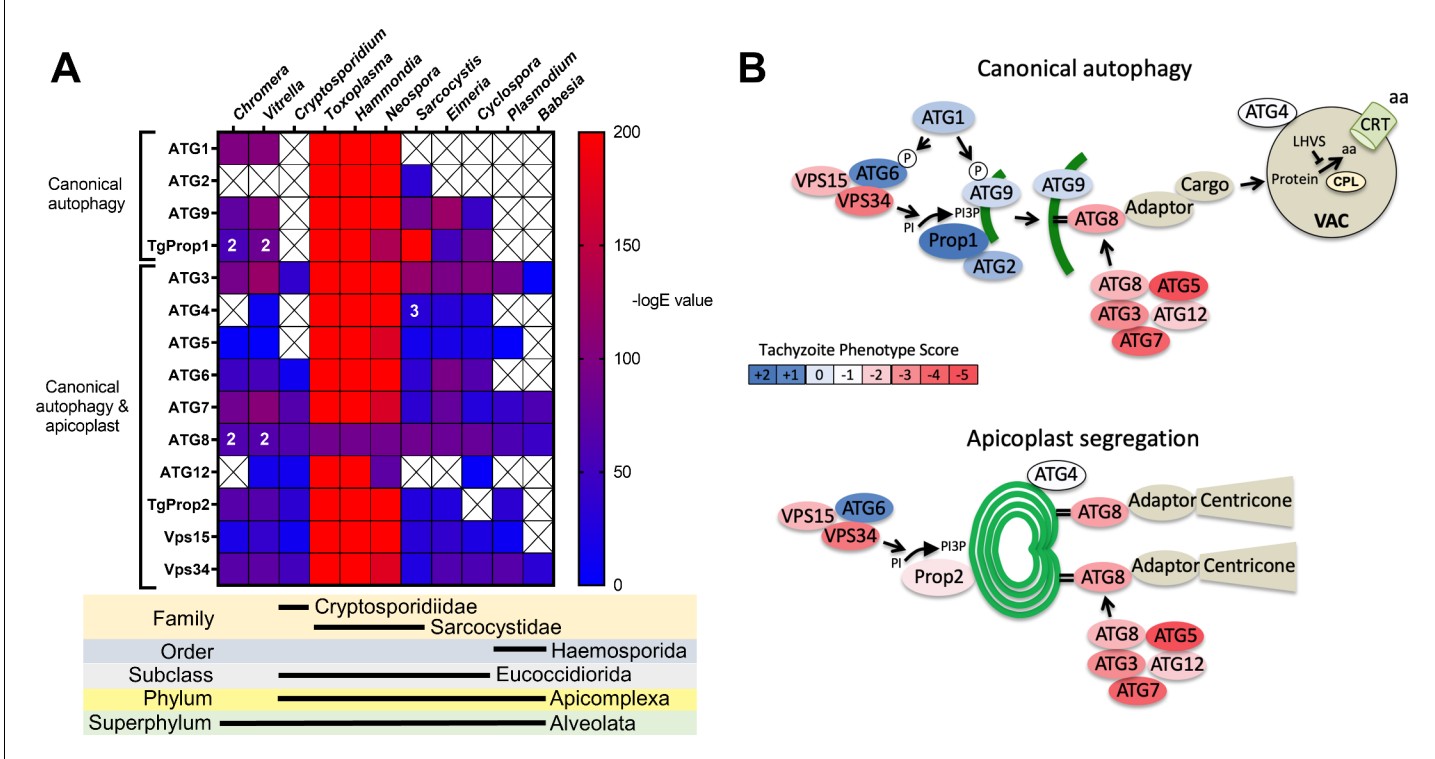

**Figure 1.** Phylogenetic conservation and adaptation of autophagy-related proteins in alveolates. (**A**) Reciprocal BLAST hits to *Toxoplasma gondii* autophagy-related proteins detected in different alveolates. The inverse logarithm of the E value (−logE) from each hit with a corresponding *T. gondii* protein is plotted in a heat map to indicate the extent to which *T. gondii* autophagy-related proteins are conserved among other alveolates. A higher −logE value indicates a closer match to *T. gondii* orthologs. The number of orthologs is shown in the corresponding cell if more than one ortholog was identified. Boxes with a cross indicate no matching ortholog was identified. (**B**) Working models of canonical autophagy and apicoplast segregation mediated by a subset of autophagy-related proteins. Autophagy-related proteins are color-coded according to phenotype scores indicated in the scale. Autophagy proteins adapted for apicoplast maintenance tend to have negative phenotype scores in tachyzoites, based on a genome-wide CRISPR/Cas9 screen in *T. gondii* (**Sidik et al., 2016**), indicating they are essential for parasite survival in the acute stage of infection. Autophagy-related proteins that have a role in canonical autophagy and have not been adapted for apicoplast maintenance tend to have neutral or positive phenotype scores, indicating canonical autophagy is a non-essential cellular pathway in acute stage parasites.

segregation in tachyzoites (*Lévêque et al., 2015*) and presumably also required for canonical autophagy, as in other systems. To achieve selective loss of expression in bradyzoites, we used CRISPR/Cas9 to insert a tachyzoite stage-specific SAG1 promoter upstream of the endogenous *TgATG8* gene in the Prugniaud (Pru) strain, thus generating S/ATG8 transgenic parasites (*Figure 2A,B*). At the same time, we also added an N-terminal GFP tag to TgATG8. S/ATG8 tachyzoites showed GFP-TgATG8 associated with the apicoplast (*Figure 2C*), which appeared to segregate normally, consistent with GFP-TgATG8 expression being sufficient to support its critical role in maintaining this organelle (*Lévêque et al., 2015*). Upon in vitro differentiation, expression of GFP-TgATG8 decreased to below the limits of detection by fluorescence microscopy and western blotting (*Figure 2C,D*). To assess the role of TgATG8 in bradyzoite autophagy, we converted S/ATG8 to bradyzoites for 7 days and looked for accumulation of dark puncta and CytoID-positive structures following treatment with the cathepsin L inhibitor LHVS. CytoID is a fluorescent dye that accumulates in autolysosomes and stains autophagic material in the VAC (*Di Cristina et al., 2017*). LHVS irreversibly blocks the activity of TgCPL resulting in the accumulation of autophagic material in the VAC, which is visible as dark puncta by phase contrast microscopy. Whereas 24 hr of LHVS treatment resulted in the expected accumulation of dark puncta and CytoID-positive structures in Pru bradyzoites, such structures were lacking in S/ATG8 bradyzoites (*Figure 2E*), hinting autophagy is indeed affected in this mutant. Dark puncta and CytoID-positive structures were even more pronounced in Pru bradyzoites following 7 days of continual LHVS treatment. Some small puncta and CytoID signal were present in S/ATG8 bradyzoites after 7 days of LHVS treatment, suggesting a measure of

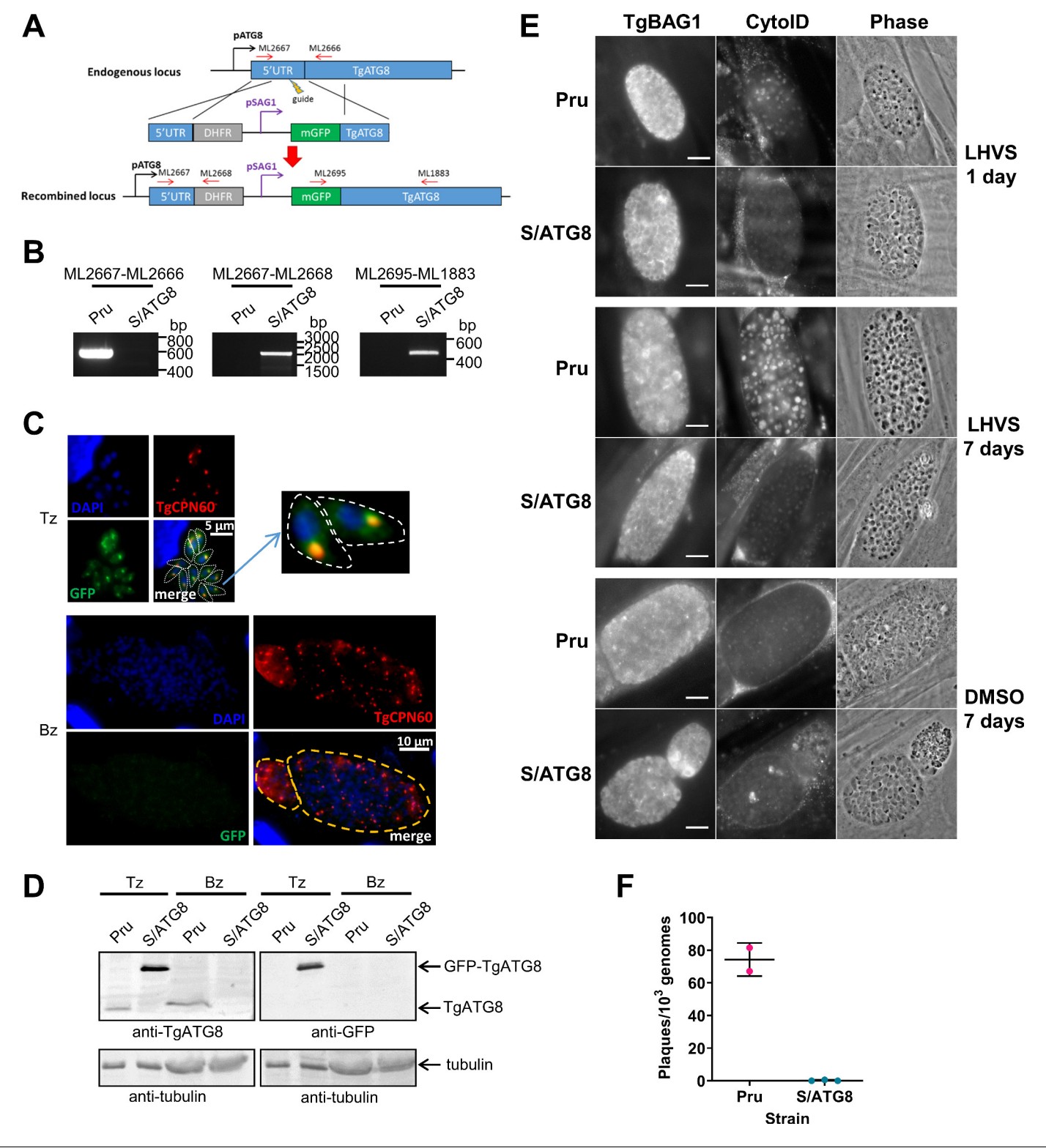

**Figure 2.** TgATG8 is required for efficient autophagosome production in *Toxoplasma gondii* bradyzoites. (**A**) Stage-specific expression and fluorescent tagging of TgATG8 was achieved by insertion of the tachyzoite-specific *TgSAG1* promoter and a GFP-coding sequence immediately upstream of the *TgATG8* gene. Site-specific insertion of the DHFR-pSAG1-mGFP repair template was performed by homology-directed repair. (**B**) Polymerase chain reaction (PCR) analysis confirmed the repair template had been inserted at the correct locus to generate a *T. gondii* strain in which *TgATG8* expression was driven by the *TgSAG1* promoter (*S/ATG8*). Stage-specific expression of GFP-TgATG8 in tachyzoites (Tz) but not bradyzoites (Bz) was

*Figure 2 continued on next page*

Figure 2 continued

confirmed by immunofluorescence microscopy (C) and western blotting (D). GFP-TgATG8 signal overlapped with the apicoplast marker TgCPN60 (a native luminal apicoplast chaperone), indicating correct localization to the apicoplast. Dashed lines outline individual parasites in the tachyzoite images and the limits of cysts on the bradyzoite images. (E) Fluorescent staining of in vitro differentiated cysts with the bradyzoite specific marker TgBAG1 and the autolysosome detection reagent CytoID. Phase contrast images indicate the development of dark puncta in LHVS-treated bradyzoites. Scale bar, 10 μm. (F) Viability of PruΔhxg and S/ATG8 bradyzoites isolated after 28 days of differentiation, as indicated by their application to fresh monolayers for plaque formation.

residual autophagy in the nominal absence of TgATG8 (*Figure 2E*). No viable S/ATG8 parasites were recovered from differentiated cultures upon isolating bradyzoites and applying them to fresh monolayers for plaque formation (*Figure 2F*). Although these experiments do not distinguish whether loss of viability is due to TgATG8's role in canonical autophagy or apicoplast segregation, the findings provide an initial indication of an active autophagy pathway in bradyzoites.

## Autophagy is a source of material for degradation in the VAC during chronic infection

To exclusively assess the role of canonical autophagy in *T. gondii* bradyzoites, it was necessary to focus on an autophagy-related protein that is required for the formation of autophagosomes but is not necessary for apicoplast maintenance. Working with a non-cystogenic strain, we previously showed that the autophagy-related protein TgATG9 was dispensable in cultured tachyzoites and that TgATG9 ablation had no observable impact on apicoplast homeostasis (*Nguyen et al., 2017*). However, TgATG9-deficient parasites showed reduced protein turnover as extracellular tachyzoites, implying a role for TgATG9 in autophagy (*Nguyen et al., 2017*). Therefore, we knocked out TgATG9 in a PruΔhxg background strain to generate a cystogenic Δatg9 strain (*Figure 3—figure supplement 1*). Genetic complementation was achieved by expressing *TgATG9* from its cognate promoter and 5' and 3' untranslated regions together with three copies of a C-terminal HA epitope tag, thus creating Δatg9ATG9. Δatg9ATG9 tachyzoites expressed *TgATG9* transcripts at levels comparable to those of PruΔhxg based on real-time quantitative PCR (qRT-PCR) (*Figure 3—figure supplement 1*), and expression of TgATG9-3×HA was confirmed by immunofluorescence microscopy and western blotting (*Figure 3—figure supplement 1*).

Using Δatg9 and Δatg9ATG9 transgenic strains, we next sought to confirm whether canonical autophagy is a source of material for degradation in the bradyzoite VAC. Bradyzoites were differentiated for 7 days in vitro and treated with either DMSO (vehicle control) or LHVS for 24 hr under differentiation conditions. Dark puncta were visible in LHVS-treated PruΔhxg and Δatg9ATG9 bradyzoites but were less prominent in Δatg9 bradyzoites (*Figure 3A*). Although a significant increase in puncta size was seen in Δatg9 bradyzoites treated with LHVS compared to the same strain treated with DMSO, this increase was ~64% lower than that for LHVS-treated PruΔhxg and ~73% lower than that for Δatg9ATG9 bradyzoites (*Figure 3B*). Also, whereas PruΔhxg and Δatg9ATG9 bradyzoites displayed CytoID-positive structures following LHVS treatment, such structures were much less prominent in Δatg9 bradyzoites (*Figure 3C*). Quantification revealed that LHVS-treated Δatg9 bradyzoites have smaller and, importantly, fewer CytoID-positive structures than those of PruΔhxg and Δatg9ATG9 bradyzoites (*Figure 3D–E*). Taken together, these findings establish that expression of TgATG9 is necessary for efficient delivery of autophagic material to the VAC in chronic stage *T. gondii*, thus suggesting a central role for TgATG9 in canonical autophagy.

## Canonical autophagy is required for normal bradyzoite morphology and cell division

To initially determine whether removal of the *TgATG9* gene had an effect on cyst development and bradyzoite morphology, we measured cyst size based on staining the cyst wall with Dolichos lectin and assessed the overall appearance of bradyzoites with the peripheral marker TgIMC1. Interestingly, we found that Δatg9 cysts are moderately larger than PruΔhxg or Δatg9ATG9 cysts when measured after 7 days of differentiation (*Figure 4A*). Individual Δatg9 bradyzoites appeared to be misshapen and bloated within their cysts, suggesting a possible basis for cyst enlargement (*Figure 4B*). When viewed by transmission electron microscopy (TEM), PruΔhxg bradyzoites showed normal ultrastructural features and typical formation of daughter cells within mother bradyzoites

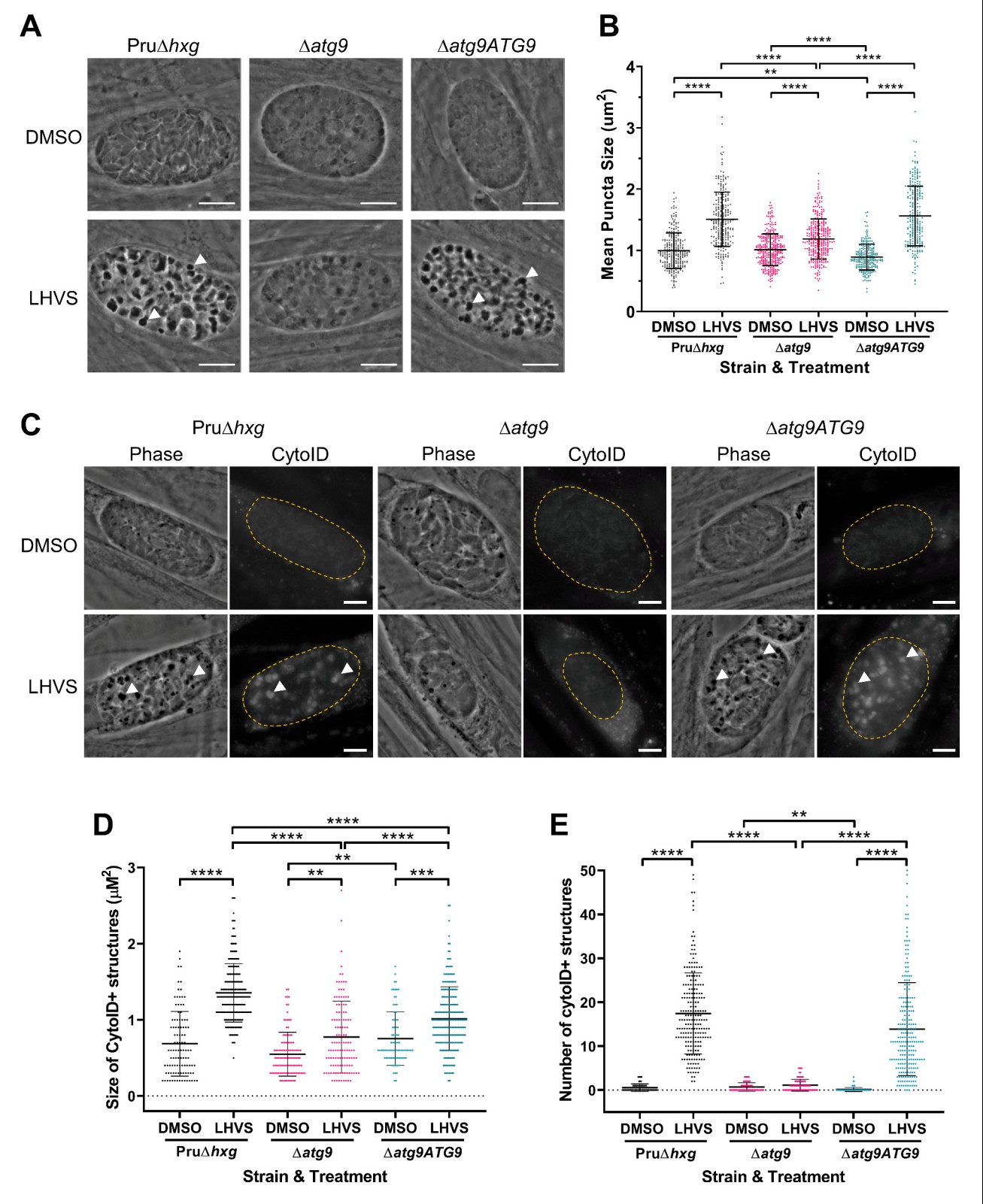

**Figure 3.** TgATG9 is required for efficient autophagosome production in *T. gondii* bradyzoites. (**A**) Phase contrast images showing that the development of dark puncta in in vitro differentiated bradyzoites following LHVS treatment is dependent on the expression of TgATG9. Arrowheads indicate a subset of dark puncta. Scale bar, 10 μm. (**B**) Quantification of dark puncta size. Each dot represents the mean size of puncta within one cyst. Data are merged from six biological replicates, with a minimum of 22 cysts analyzed per sample type, per biological replicate, and a minimum total of

*Figure 3 continued on next page*

Figure 3 continued

242 cysts analyzed per sample type across all biological replicates. (C) CytoID staining of autolysosomes in in vitro differentiated bradyzoite cysts. Arrowheads indicate an arbitrary subset of dark puncta and the corresponding CytoID-positive structures. Scale bar, 5 µm. Quantification of the size (D) and number (E) of CytoID-positive structures in in vitro differentiated bradyzoites. Each dot represents the average puncta measurement within a single cyst. Data are merged from three biological replicates, with a minimum total of 81 cysts analyzed per sample type across three biological replicates. Cysts were identified by Dolichos lectin staining and measurements of dark puncta and CytoID were made exclusively within Dolichos lectin-positive regions. For panels B, D, and E, bars represent mean ± SD. Statistical comparisons were done using a Kruskal-Wallis test with Dunn's multiple comparisons. Statistical significance is indicated as follows: **p<0.01; ***p<0.001; ****p<0.0001. Non-significant differences are not indicated. Statistical analysis is only shown for comparisons that have one variable, that is, different strain or different treatment.

The online version of this article includes the following figure supplement(s) for figure 3:

**Figure supplement 1.** Targeted deletion and genetic complementation of TgATG9.

---

(*Figure 4Ca,Da*). By contrast, Δ*atg9* bradyzoites were often deformed, vacuolized, and showed abnormal ultrastructural features including an enlarged nuclear envelope (*Figure 4Cb*). Some Δ*atg9* bradyzoites also showed aberrant formation or budding of daughter cells, which manifested as enlarged parasites containing multiple daughter nuclei (*Figure 4Db,Dc*). This multinucleated phenotype was found in 26% (61/235) of Δ*atg9* bradyzoites but less than 1% (2/144) of PruΔ*hxg* bradyzoites (*Table 1*). Together, these findings suggest that TgATG9 is necessary for the overall fitness of bradyzoites and its loss compromises normal bradyzoite morphology and cell division.

## A lack of canonical autophagy leads to ultrastructural abnormalities in the VAC of bradyzoites

The VAC performs a lysosome-like function in *T. gondii* tachyzoites and bradyzoites and is the site for proteolytic turnover of exogenously and endogenously derived proteinaceous material, including autophagosomes (*Dou et al., 2014*; *McGovern et al., 2018*; *Smith et al., 2020*; *Di Cristina et al., 2017*). Compared to the normal small and mixed electron density appearance of the VAC in PruΔ*hxg* bradyzoites (*Figure 5Aa*), TEM imaging revealed various abnormalities for this organelle in Δ*atg9* bradyzoites including some VACs that appeared enlarged and empty (*Figure 5Ab*). More commonly, other Δ*atg9* VACs were filled with highly electron-dense material (*Figure 5Ac*), which sometimes included organellar remnants (*Figure 5Ad*). Whereas no PruΔ*hxg* (DMSO) bradyzoites had an electron-dense VAC, 12.3% of Δ*atg9* bradyzoites showed this phenotype, suggesting a link between TgATG9 and proteolytic digestion in the VAC. The observation of electron-dense material in Δ*atg9* bradyzoites (DMSO) is similar to, albeit much lower than LHVS-treated PruΔ*hxg* bradyzoites (*Figure 5Ba*, b), for which 93.1% showed an electron-dense VAC. The VAC of LHVS-treated Δ*atg9* bradyzoites was sometimes only partially electron dense (*Figure 5Bc*), but the great majority (94.2%) of such parasites showed electron-dense material (*Figure 5Bd*). This is consistent with our earlier observation that LHVS treatment increases the size of dark puncta in Δ*atg9* bradyzoites, albeit not to the same degree as LHVS-treated PruΔ*hxg* or Δ*atg9ATG9* bradyzoites.

## Multimembrane structures and abnormal mitochondria in Δ*atg9* bradyzoites

Autophagic structures are typically transient and thus are rarely observed unless autophagic flux (the production and turnover of autophagosomes) is affected by increased biogenesis, reduced degradation, or both. This property is consistent with our inability to see autophagic structures by TEM in vehicle-treated PruΔ*hxg* (DMSO) bradyzoites, whereas 6.9% of PruΔ*hxg* bradyzoites treated with LHVS harbored such structures (*Table 1*). Interestingly, 6% of vehicle-treated and 10.1% of LHVS-treated Δ*atg9* bradyzoites also showed vesicular structures displaying two or more membranes, as exemplified in *Figure 6A*. Although it is uncertain whether these structures are autophagosomal, these observations suggest that TgATG9 is not required for the biogenesis of these multimembrane structures, and that ablation of TgATG9 potentially reduces the rate at which vesicular material is turned over by the VAC.

In addition to the other abnormalities described above, the mitochondria of Δ*atg9* bradyzoites showed strikingly aberrant appearance. Whereas the mitochondria in PruΔ*hxg* bradyzoites displayed their typical tubular network, the mitochondria of Δ*atg9* bradyzoites appeared fragmented and punctate when observed by immunofluorescence assay (IFA) (*Figure 6H*). Closer inspection by TEM

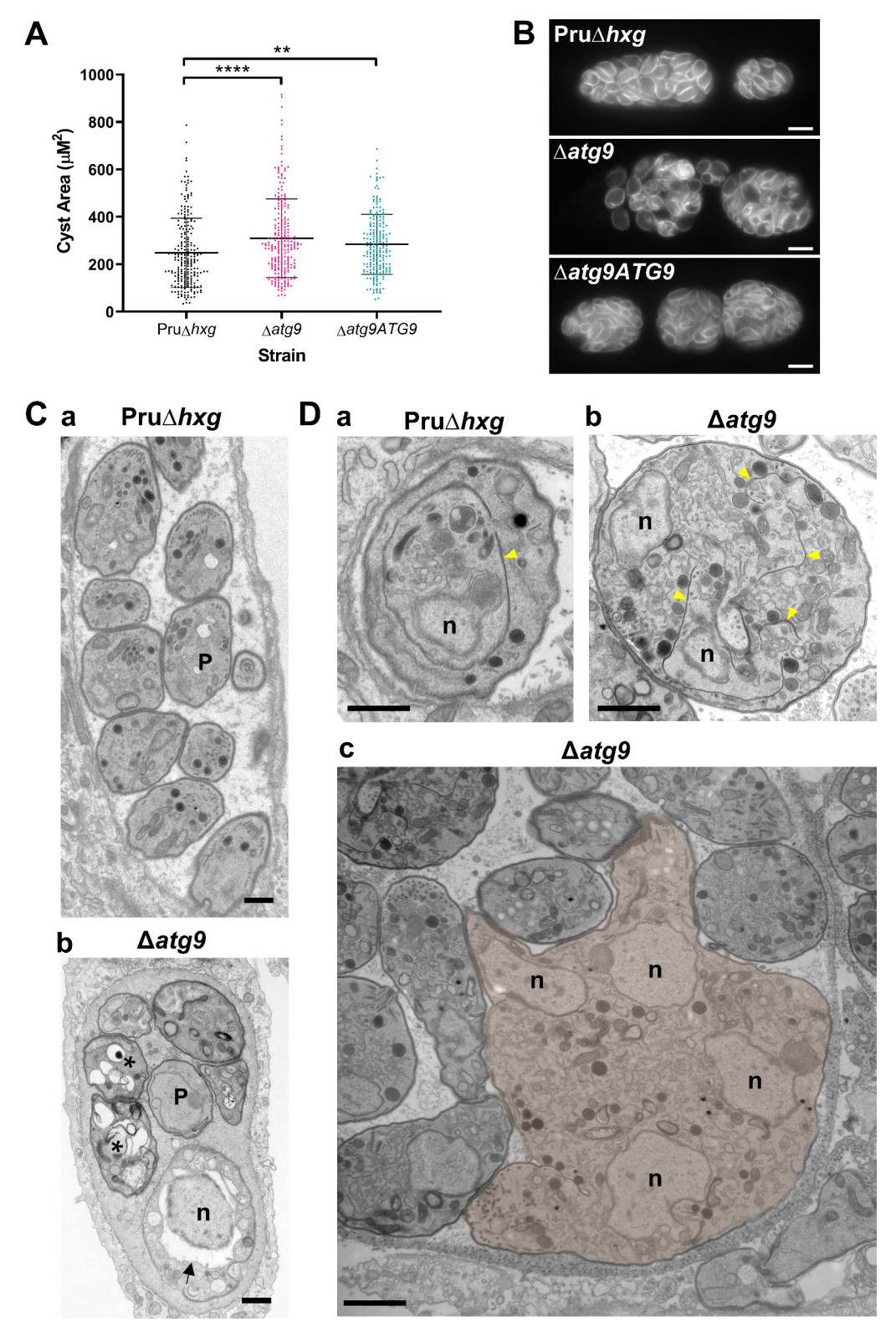

**Figure 4.** Canonical autophagy is required for normal bradyzoite morphology and cell division. (A) Quantification of cyst size based on staining for the cyst wall with Dolichos lectin. Each dot represents the size of one cyst. Data are merged from three biological replicates, each with a minimum of 52 cysts analyzed per sample type, per biological replicate, and a minimum total of 199 cysts analyzed per sample type across three biological replicates. Bars indicate mean ± SD. Statistical comparisons were done using a Kruskal-Wallis test with Dunn's multiple comparisons. Significance is indicated as

*Figure 4 continued on next page*

Figure 4 continued

**p<0.01; ****p<0.0001. Non-significant differences are not indicated. (**B**) Immunofluorescence imaging of bradyzoites stained with anti-TgIMC1 revealed disorganization of the inner membrane complex in Δatg9 parasites and a 'bloating' phenotype. Scale bar, 10 μm. (**C**) Transmission electron microscopy (TEM) showing a cyst containing normally developed bradyzoites from PruΔhxg (a) and aberrant Δatg9 bradyzoites (b), with highly vacuolized, dying parasites (asterisks) or abnormally enlarged nuclear envelope (arrow). P, parasite. (**D**) TEM showing normal endodyogeny for PruΔhxg bradyzoites with a nascent daughter containing well-organized organelles surrounded by the IMC (yellow arrowhead in panel a), and aberrant parasite division with poor packaging of organelles by the IMC (b), resulting in enlarged multinucleated Δatg9 bradyzoites (pseudocolored parasite, c). Note that PruΔhxg and mutant parasites are shown at the same magnification for direct comparison. All scales bars are 1 μm.

confirmed that while the mitochondria of PruΔhxg bradyzoites appeared normal, with internal cristae throughout (*Figure 6B*), 22.6% of Δatg9 bradyzoites had extremely thin mitochondria that presented a horseshoe-like appearance with bulbous ends, wherein cristae were present (*Figure 6C*). Often, the two ends of the U-shaped structure appeared to fuse together, encapsulating components of the cytoplasm, including ER, secretory organelles (dense granules and micronemes), lipid droplets, and even other aspects of the mitochondria itself (*Figure 6D*). That such structures were comprised of four membranes (e.g., *Figure 6E* inset) clearly distinguished them from autophagosomes and the abnormal double-membrane vesicles observed. LHVS treatment of Δatg9 bradyzoites did not seem to change the appearance or frequency (23.7%) of the abnormal mitochondria (*Figure 6E*). These thin mitochondria were never seen in vehicle-treated PruΔhxg bradyzoites, but they were observed after LHVS treatment at a frequency of 11.9%.

In order to quantitatively assess changes in mitochondria morphology, CellProfiler was used to define the architecture of anti-$F_1F_0$ATPase-labeled parasite mitochondria within Dolichos lectin-labeled bradyzoite cysts, based on various mitochondria shape-based parameters. The dimensionality of the mitochondria morphology data was then reduced using principal component analysis, providing an overall relative $F_1F_0$ATPase score (a.u.). Relative $F_1F_0$ATPase scores were only significantly different between mutant Δatg9 parasites and the parental (PruΔhxg) and complemented (Δatg9ATG9) parasites. Furthermore, these differences were only recorded in parasites 6 (PruΔhxg vs. Δatg9, p=0.004) and 8 (PruΔhxg vs. Δatg9, p=0.0026; Δatg9 vs. Δatg9ATG9, p=0.001) days after the initiation of in vitro differentiation from tachyzoites to bradyzoites (*Figure 6F,G*). No significant differences were observed 1 or 3 days after the initiation of in vitro differentiation (*Figure 6F*). Taken together, these findings suggest that impairment of autophagic flux due to ablation of TgATG9 or inhibition with LHVS leads to striking mitochondrial abnormalities in bradyzoites.

## Canonical autophagy is critical for bradyzoite viability and persistence

Observations by electron microscopy suggested that 30% of Δatg9 cysts showed signs of degeneration (*Table 1*), implying that a functional autophagy pathway is important for bradyzoite survival. To more directly assess the role of TgATG9 in bradyzoite viability, we differentiated PruΔhxg, Δatg9, and Δatg9ATG9 for 7 days in vitro before treating them with vehicle or LHVS for 7 or 14 days under continued differentiation conditions and assessing viability by quantitative polymerase

**Table 1.** Phenotypes observed by transmission electron microscopy (TEM).

| Phenotype | PruΔhxg | | Δatg9 | |
| --- | --- | --- | --- | --- |
| | **DMSO** | **LHVS** | **DMSO** | **LHVS** |
| Abnormally dividing bradyzoites | <1 (2/144)[1] | 8.4 (17/202) | 26.0 (61/235) | 23.2 (44/190) |
| Live[2] bradyzoites with electron-dense vacuolar compartment (VAC) | 0 (0/144) | 93.1 (188/202) | 12.3 (29/235) | 94.2 (179/190) |
| Autophagic structures with two membranes in live bradyzoites | 0 (0/144) | 6.9 (14/202) | 6.0 (14/235) | 10.5 (20/190) |
| Abnormally thin mitochondria in live bradyzoites | 0 (0/144) | 11.9 (24/202) | 22.6 (53/235) | 23.7 (45/190) |
| Degenerated cysts | 3.0 (1/33) | 11.8 (6/51) | 30.0 (18/60) | 23.4 (11/47) |

[1]Values are percentage (number of bradyzoites or cysts with phenotype/total number of bradyzoites or cysts enumerated).

[2]Only bradyzoites that appeared to be intact and likely viable were included in the analysis.

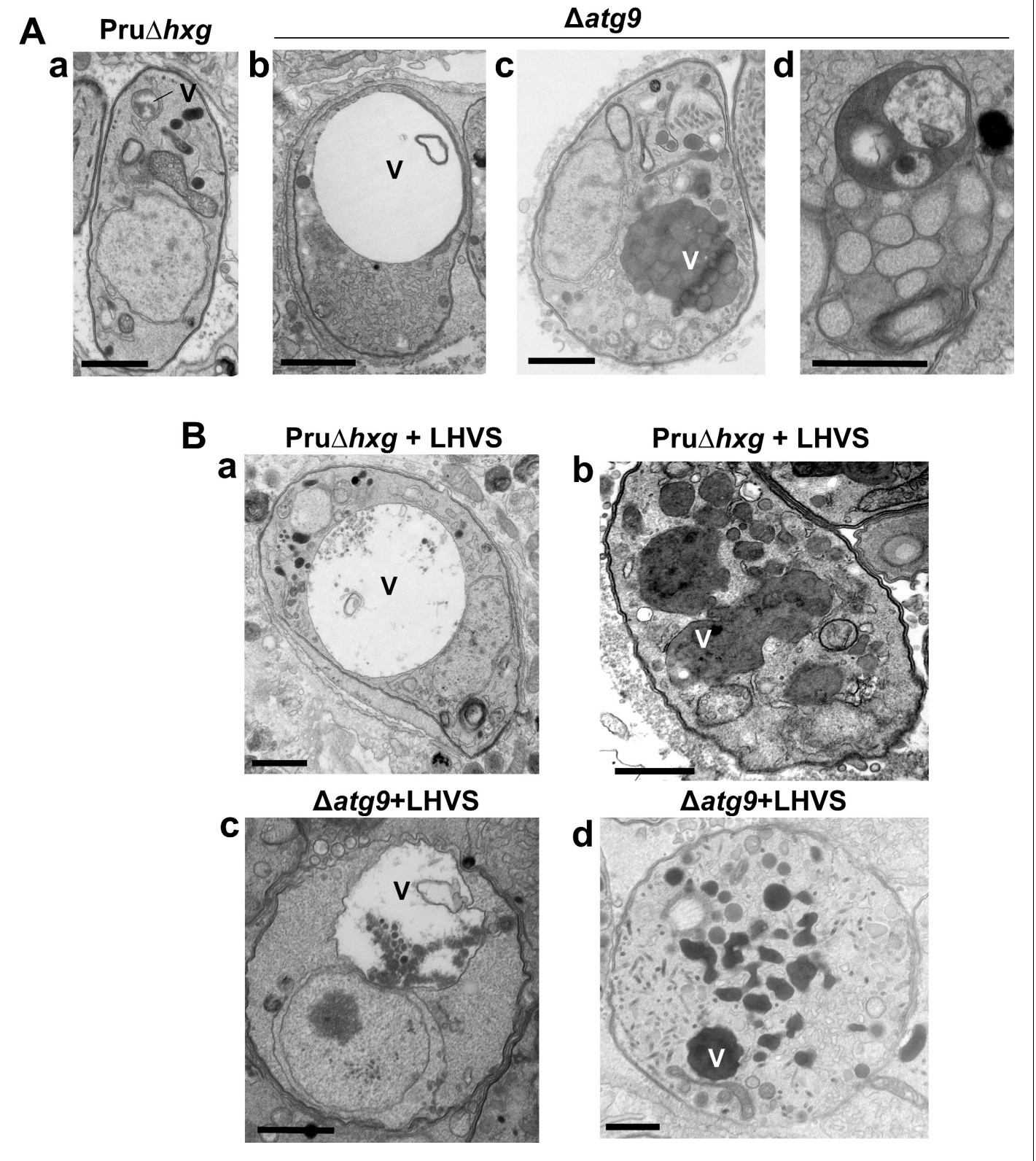

**Figure 5.** A lack of canonical autophagy leads to ultrastructural abnormalities in the vacuolar compartment (VAC) of bradzoites. (**A**) Transmission electron microscopy (TEM) showing a normal VAC (V) in PruΔ*hxg* bradzoites (panel a). Three types of VAC abnormalities were observed in Δ*atg9* bradzoites: a very enlarged electron-lucent (e-lucent) compartment (panel b), small electron-dense vesicles (panel c), or a very enlarged electron-dense

*Figure 5 continued on next page*

*Figure 5 continued*

compartment (panel d). (B) TEM of VAC in PruΔ*hxg* and Δ*atg9* bradyzoites treated with LHVS showing for both strains very large VAC with e-lucent content (panels a and c) or electron-dense content (panels b and d). All scales bars are 1 μm.

chain reaction (qPCR)/plaque assay. Consistent with previous studies (*Di Cristina et al., 2017*), LHVS treatment compromised bradyzoite viability, particularly after 14 days of treatment (*Figure 7A,B*). As expected, PruΔ*hxg* and Δ*atg9ATG9* bradyzoites treated with DMSO remained viable at both time points. However, vehicle-treated Δ*atg9* bradyzoites were much less viable than Δ*atg9ATG9* at both 7 and 14 days of treatment (corresponding to 14 and 21 days of differentiation). Δ*atg9* bradyzoites treated with LHVS showed a trend toward lower viability compared to LHVS-treated PruΔ*hxg* and Δ*atg9ATG9*. We repeated the experiment to perform statistical comparisons of viability after 14 days of differentiation without treatment. The findings confirmed that Δ*atg9* bradyzoites are significantly less viable than PruΔ*hxg* or Δ*atg9ATG9* bradyzoites (*Figure 7C*). To distinguish whether the reduced viability in Δ*atg9* bradyzoites was a result of a differentiation or a persistence defect, in vitro bradyzoite differentiation assays were performed. We determined the relative SAG1:BAG1 signal within individual PVs/cysts at four different time points (1–4 days from the initiation of differentiation in vitro) and found there was no defect in the ability of the Δ*atg9* strain to differentiate (*Figure 7— figure supplement 1*). Conversely, relative BAG1 to SAG1 signal was found to increase more rapidly with the Δ*atg9* parasite cell line, indicating it differentiates faster than the parental strain. We also performed plaque assays on PruΔ*hxg*, Δ*atg9*, and Δ*atg9ATG9* tachyzoites and found each strain formed plaques equally well, with no significant differences in the number of plaques formed between each strain (*Figure 7—figure supplement 2*). This indicates that TgATG9 is non-essential in tachyzoites. It also suggests that host cell invasion is not measurably impacted in parasites lacking TgATG9 (*Figure 7—figure supplement 2*) and that the lower plaque number recorded in Δ*atg9* bradyzoites is due to reduced parasite viability within the cyst, as opposed to impaired host cell invasion as parasites were placed onto a new monolayer for plaquing.

To determine whether canonical autophagy is necessary for the persistence of *T. gondii* tissue cysts in vivo, we infected mice with PruΔ*hxg*, Δ*atg9*, and Δ*atg9ATG9* tachyzoites. Consistent with a previous study (*Nguyen et al., 2017*), we found that TgATG9 is necessary for normal virulence. Whereas 30% (6/20) and 20% (2/10) of mice infected with PruΔ*hxg* or Δ*atg9ATG9*, respectively, failed to survive to 35 days (5 weeks) post-infection, all mice (10/10) infected with the Δ*atg9* strain survived (*Figure 7D*). qPCR analysis of brain homogenate during the acute stage at day 7 post-infection showed that initial parasite brain burden did not differ significantly between mice infected with PruΔ*hxg*, Δ*atg9*, and Δ*atg9ATG9* (*Figure 7E*), suggesting that TgATG9 deficiency does not overtly affect parasite dissemination to the brain. By contrast, in mice that survived to 5 weeks post-infection, we observed an ~100-fold decrease in the brain cyst burden for mice infected with Δ*atg9* compared to those infected with PruΔ*hxg* or Δ*atg9ATG9* (*Figure 7F*). Taken together, our findings suggest that canonical autophagy is critical for *T. gondii* bradyzoite long-term viability and persistence within tissue cysts in vivo.

## Discussion

Autophagy is a complex intracellular pathway that facilitates the removal of endogenous proteins and organelles in eukaryotic cells. Often regarded as a starvation response to recycle amino acids for the production of new proteins, this process can also serve to remove misfolded proteins and damaged organelles, 'reshape' the cell during development, and, for mammalian cells, to remove pathogens (xenophagy) (*Yin et al., 2016*). In apicomplexan parasites, many autophagy-related proteins have been repurposed to have an additional role in maintaining the apicoplast, a remnant organelle derived from phagocytosis of an algal ancestor that is essential for parasite viability (*Besteiro, 2017*; *Lévêque et al., 2015*; *Lévêque and Besteiro, 2016*; *Lévêque et al., 2016*). Many of the autophagy-related proteins essential in this process, such as TgATG8, typically function 'downstream' in the autophagy pathway (as depicted in *Figure 1B*). While a number of proteins involved in apicoplast maintenance have been studied and shown to be essential in *T. gondii* and Plasmodium spp. (*Besteiro, 2017*; *Nguyen et al., 2018*; *Lévêque et al., 2015*; *Lévêque and Besteiro, 2016*;

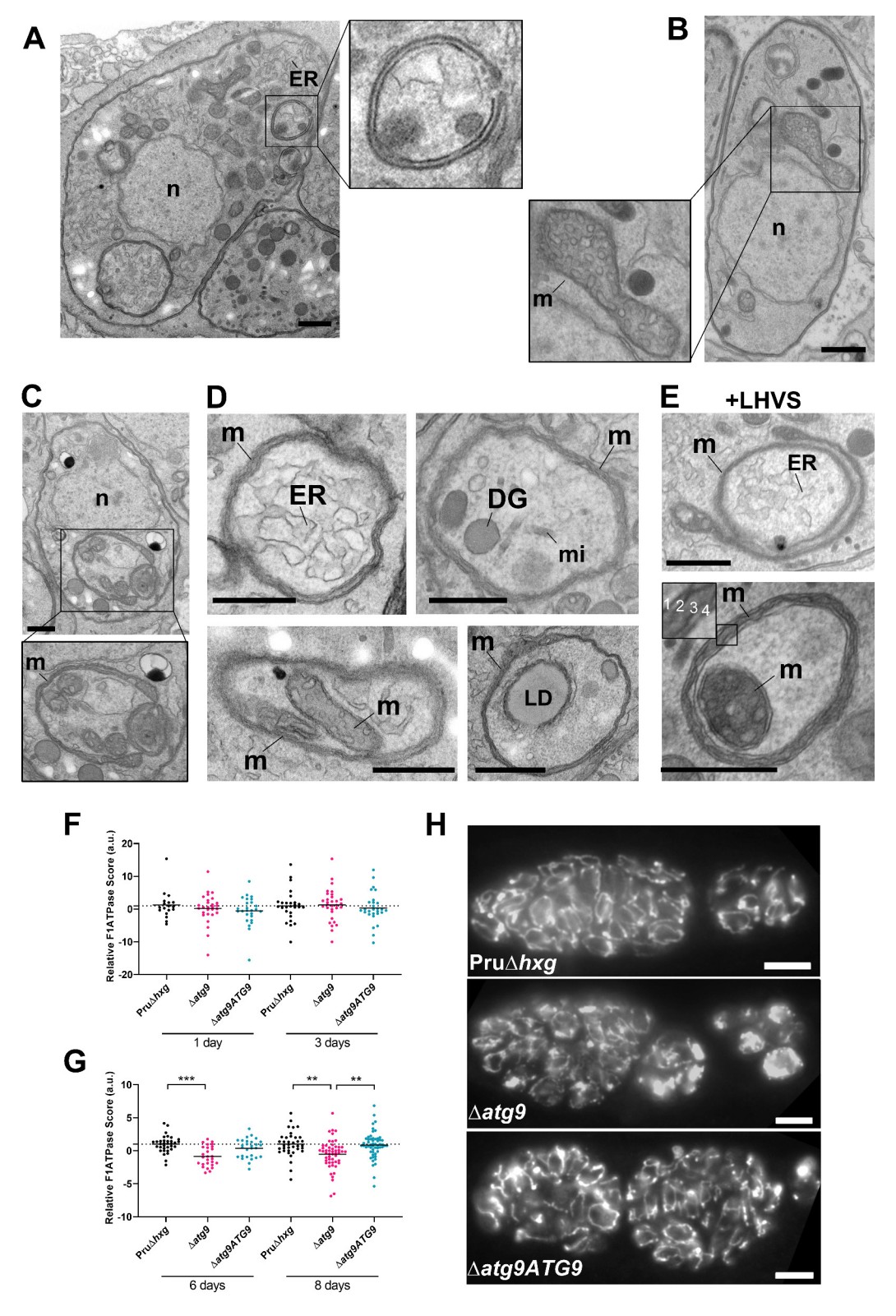

**Figure 6.** Autophagic structures and abnormal mitochondria in Δ*atg9* bradyzoites. (**A**) Transmission electron microscopy (TEM) of Δ*atg9* parasites showing a double-membrane structure containing cytoplasmic components including endoplasmic reticulum (ER). (**B**) TEM showing a normal mitochondrion in PruΔ*hxg* bradyzoites. (**C**) TEM of Δ*atg9* bradyzoites presenting thin mitochondria (m) in a horseshoe conformation with bulbous ends, wherein cristae can be seen. Nucleus (n) is denoted. (**D**) TEM of Δ*atg9* parasites revealing mitochondrial profiles (m) enveloping many organelles,

*Figure 6 continued*

including ER, dense granules (DG), micronemes (mi), mitochondria (m), and lipid droplets (LD). (E) Examples of ER and mitochondrial section wrapped by the mitochondrion after bradyzoite treatment with LHVS. Inset of the lower panel illustrates the four membranes observed in such structures. All TEM scale bars are 500 nm. Mitochondria morphology scores (relative $F_1F_0$ATPase score) for PruΔ*hxg*, Δ*atg9*, and Δ*atg9ATG9* parasites 1 and 3 (F) or 6 and 8 (G) days after the addition of alkaline media to stimulate differentiation from the tachyzoite to bradyzoite life stage. Relative $F_1F_0$ ATPase Score was normalized to the PruΔ*hxg* strain within each time point. Each data point represents one cyst. Solid lines represent the mean from three biological replicates. Statistical significance between samples was determined using a Kruskal-Wallis test with Dunn's multiple comparisons and is indicated with **p<0.01 or ***p<0.001. (H) Immunofluorescence imaging of bradyzoites stained with α-$F_1F_0$ ATPase revealed an aberrant mitochondrial network in some Δ*atg9* parasites. Scale bar, 10 μm.

*Lévêque et al., 2016*; *Jayabalasingham et al., 2014*; *Voss et al., 2016*; *Besteiro et al., 2011*), little is known about the role of proteins that function earlier in the pathway.

Until recently, the precise mechanism of the autophagy-related protein ATG9 and the expansion of the phagophore lipid bilayer remained unknown. However, recent seminal papers on this topic have revealed that human and yeast ATG9 are scramblases embedded within the lipid bilayer of autophagosomes that translocates different phospholipids between the inner and outer leaflets of the membrane (*Maeda et al., 2020*; *Orii et al., 2021*; *Matoba et al., 2020*; *Matoba and Noda, 2020*; *Guardia et al., 2020*). Structural analysis revealed ATG9 forms a homotrimer containing three lateral pores that lead into a vertical pore and mutations in either of these channels precluded phospholipid translocation and prevented the formation of complete autophagosomes (*Maeda et al., 2020*; *Matoba et al., 2020*; *Guardia et al., 2020*). Moreover, following recruitment to the phagophore assembly site by the ATG1 complex, ATG9-embedded vesicles have been described as an anchoring point for autophagosome nucleation and the recruitment of other autophagy-related proteins to the developing membrane (*Sawa-Makarska et al., 2020*; *Rao et al., 2016*). ATG9 is therefore an integral protein in the formation of autophagosomes in canonical autophagy and deletion of *ATG9* prevents the formation of autophagosomes. Whether TgATG9 also functions as a scramblase in *T. gondii* remains to be determined and will be the focus of future work.

A cathepsin L enzyme in *T. gondii*, TgCPL, has been shown to be a major protease required for the turnover of exogenous and endogenous proteinaceous material in the lysosome-like organelle, termed the VAC (*Dou et al., 2014*; *Di Cristina et al., 2017*). Similar to TgATG9, disruption of TgCPL has been found to be non-essential for tachyzoite growth and replication within the PV in vitro (*Ortega, 2019*; *Pfefferkorn et al., 1986*). This is likely due to parasites having sufficient access to nutrients (e.g., amino acids) from the host cell cytoplasm to support their rapid growth (*Dou et al., 2014*). Conversely, ablation of TgCPL function is catastrophic in chronic-stage bradyzoites, with TgCPL-deficient bradyzoites failing to survive in vitro and in vivo (*Smith et al., 2020*; *Di Cristina et al., 2017*). Accumulation of autophagic material in the VAC of TgCPL-deficient bradyzoites coincided with loss of viability (*Smith et al., 2020*; *Di Cristina et al., 2017*). This presented the possibility that although canonical autophagy had been found to be dispensable in intracellular tachyzoites (*Nguyen et al., 2017*), it could be an important pathway in chronic-stage bradyzoites. Here, we demonstrated that removal of the *TgATG9* gene substantially reduces the accumulation of autophagic material in bradyzoites treated with LHVS to inhibit the enzymatic activity of TgCPL. Importantly, we also demonstrated that genetic ablation of *TgATG9* resulted in bradyzoite death in vitro, in both TgCPL-proficient and LHVS-treated (effectively TgCPL-deficient) parasites. When autophagic material was previously reported to accumulate in Δ*cpl T. gondii* bradyzoites (*Di Cristina et al., 2017*), it was possible that autophagy was triggered in response to starvation, resulting from a general block in TgCPL-dependent protein turnover in the VAC. However, we found that the survival of vehicle control DMSO-treated (TgCPL-proficient) Δ*atg9* bradyzoites was severely compromised in vitro, supporting the hypothesis that autophagy is a primary pathway in chronic-stage *T. gondii* which is essential for survival within the cyst. This is supported by in vivo experiments performed in mice in which Δ*atg9* brain cysts were severely reduced at 5 weeks post-infection. Having detected Δ*atg9* parasites in the brains of mice 1 week post-infection, we stipulate that autophagy is a survival mechanism in *T. gondii* that facilitates parasite persistence within the cyst. Furthermore, we found no defect in the ability of Δ*atg9* parasites to differentiate from tachyzoites to bradyzoites, indicating that reduced in vitro and in vivo bradyzoite survival was due to an inability to persist, as opposed to an inability to differentiate to the bradyzoite life stage and form cysts.

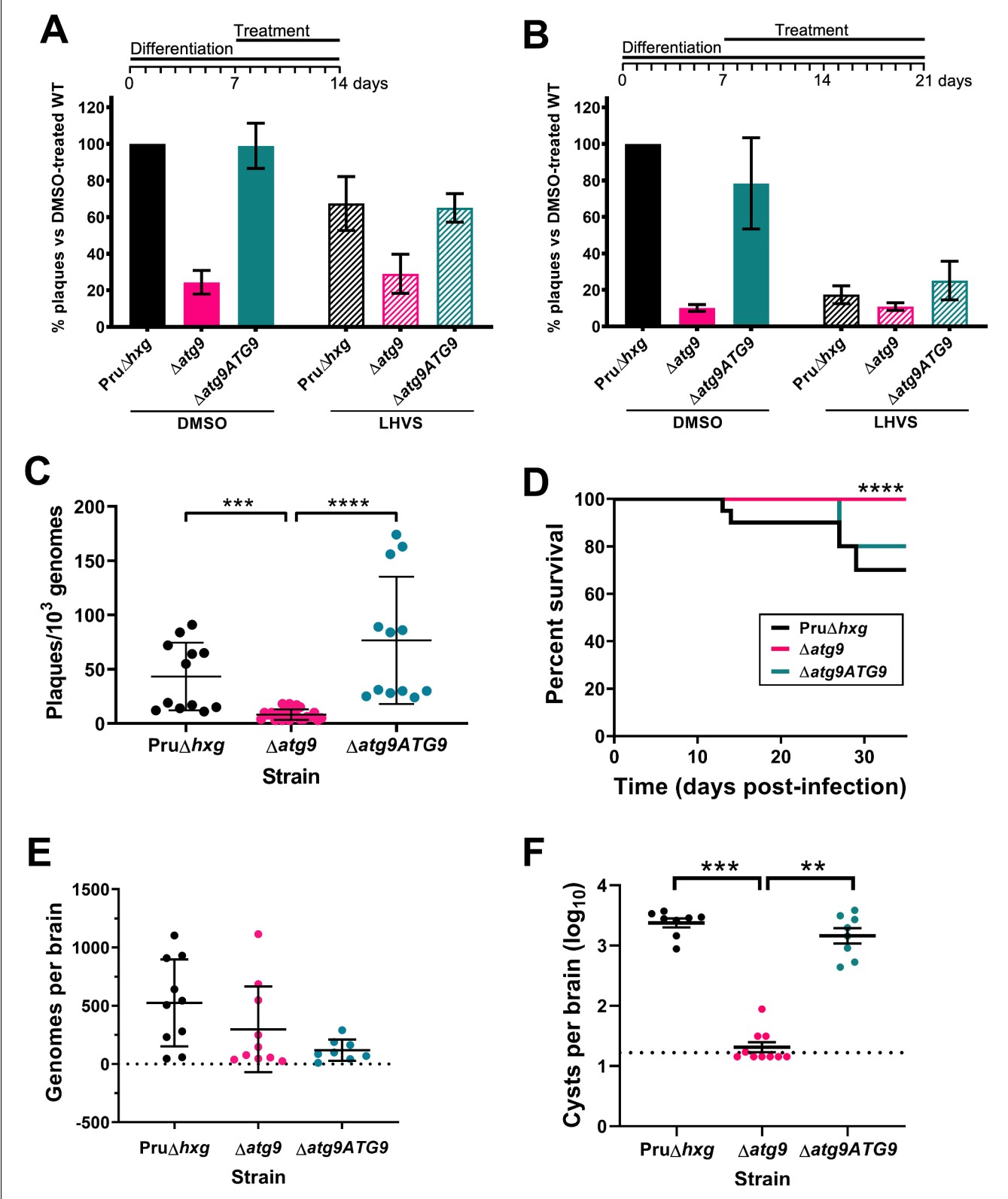

**Figure 7.** Autophagy is required for bradyzoite viability and the persistence of tissue cysts. Quantification of bradyzoite viability by quantitative polymerase chain reaction (qPCR)/plaque assay after differentiation media for 1 week followed by treatment with DMSO (vehicle control, blue) or LHVS (red) for a further 1 week (A) or 2 weeks (B). Data are from three biological replicates each with three technical replicates and are normalized to PruΔ*hxg* set at 100%. Bars represent mean ± SEM. (C) Viability of untreated bradyzoites after 2 weeks of differentiation in vitro. Each dot

*Figure 7 continued on next page*

*Figure 7 continued*

represents viability from one technical replicate. Data are merged from four biological replicates. Statistical comparisons were done using a Kruskal-Wallis test with Dunn's multiple comparisons. Significance is indicated as ***p<0.001; ****p<0.0001. Non-significant differences are not indicated. (D) Analysis of virulence based on survival of infected mice. Mice were infected intraperitoneally with 150 tachyzoites of each strain. Infected mice that became moribund before the endpoint at day 35 post-infection were humanly euthanized. Mice infected with Δatg9 showed significantly better survival based on Mantel-Cox log-rank test (****p<0.0001). (E) Analysis of parasite dissemination to the brain. Brains were harvested from infected mice at 7 days post-infection and the number of genomes present in each brain homogenate was determined by qPCR. No significant differences were seen between strains based on analysis with a Kruskal-Wallis test with Dunn's multiple comparisons. (F) Analysis of brain cyst burden in infected mice. Brains were harvested from mice 5 weeks post-infection and cysts were counted in blinded samples of brain homogenates by microscopy. Data is plotted as $\log_{10}$ cysts per brain. The dotted line intersecting the y axis represents the limit of detection (<33 cysts/brain). Samples for which no cysts were observed in the 30 µl of homogenate analyzed were given a value that is half the limit of detection. Bars represent mean ± SEM and statistical comparisons were done using a Kruskal-Wallis test with Dunn's multiple comparisons. Significance indicated as **p<0.01, ***p<0.001. Non-significant differences are not indicated.

The online version of this article includes the following figure supplement(s) for figure 7:

**Figure supplement 1.** Analysis of differentiation.
**Figure supplement 2.** TgATG9 is not required for tachyzoite plaque formation.

Upon closer inspection of in vitro bradyzoites by TEM, we noticed that treatment with LHVS led to numerous ultrastructural abnormalities in the VAC in both PruΔhxg and Δatg9. For example, enlarged 'empty' VACs were observed following LHVS treatment, which were also detected in Δatg9 bradyzoites treated with DMSO. Taken together, this suggests that TgATG9 and possibly autophagic flux play an important role in VAC homeostasis. As mentioned above, inhibition of TgCPL with LHVS predominantly results in enlargement of dark puncta, which correspond to accumulation of electron-dense and autophagic material in the VAC, shown here and previously documented by *Di Cristina et al., 2017*. We observed a less pronounced enlargement of dark puncta in LHVS-treated Δatg9 bradyzoites compared with LHVS-treated PruΔhxg or Δatg9ATG9. This, coupled with our observation that 90% of LHVS-treated Δatg9 bradyzoites have an electron-dense VAC, suggests that some trafficking of proteinaceous material to the VAC still occurs in such parasites. However, since we did not see an increase in the number of CytoID-positive puncta in LHVS-treated Δatg9 bradyzoites compared with DMSO-treated Δatg9, this proteinaceous material does not seem to be derived from canonical autophagosomes. We propose two possibilities, that bradyzoites uptake host proteins that are trafficked to the VAC for degradation via the endolysosomal system or that, in the absence of TgATG9, bradyzoites have an alternative mechanism of autophagy. The first hypothesis is supported by our recent work, which demonstrated that bradyzoites can ingest and degrade host-derived cytosolic proteins (*Kannan et al., 2021*). In the second possibility, as discussed further below, we observed the formation of vesicles with two or more membranes in Δatg9 bradyzoites by TEM and while we are currently unable to determine the nature of these vesicles, it remains possible that they represent alternative autophagosome-like structures that do not stain well with the cationic amphiphilic dye CytoID. In either case, the inability of Δatg9 bradyzoites to survive and persist suggests that neither of these alternative pathways can rescue the function of canonical autophagy in *T. gondii* bradyzoites.

TEM imaging also identified numerous structural anomalies that were present in Δatg9 bradyzoites but absent in the parental strain. Structural abnormalities included bradyzoites with a division defect, manifested by, for example, the observation of four nuclei within a single bradyzoite. Although endopolygeny has previously been reported in *T. gondii* bradyzoites (*Dzierszinski et al., 2004*), we found it to occur in less than 1% of PruΔhxg parasites. However, a sharp increase in the number of mother cells containing multiple daughter nuclei was observed in the *TgATG9* knockout strain, with ~30% of bradyzoites showing this phenotype. This finding implicates TgATG9 in bradyzoite replication and development, although whether the protein has a direct role or if this observation was an indirect consequence from a lack of canonical autophagy, is unclear. This is notably in contrast to what has previously been observed in TgATG9-deficient *T. gondii* tachyzoites, in which the parasite lytic cycle (acute stage invasion, replication, and egress) was found to be unaffected by the gene knockout (*Nguyen et al., 2017*). Other structural abnormalities observed in Δatg9 bradyzoites included a disorganization of the inner membrane complex and the occurrence of vesicular objects that had multiple membranes. In some cases, these structures had two double membranes

and could represent autophagosome-like vesicles produced in a TgATG9-independent manner. Although ATG9 is required for efficient autophagy, alternative vesicle and puncta formation has been found to occur in other eukaryotic organisms in which *ATG9* has been knocked out or silenced (*Kishi-Itakura et al., 2014*; *Jacquin et al., 2017*; *Runwal et al., 2019*).

In addition to the double-membrane vesicles described above, we also observed structures with at least four membranes that appeared to be derived from the mitochondria, with cristae clearly visible. A primary role of ATG9 in autophagy is to initiate the formation of the lipid bilayer for the developing phagophore (*Zhou et al., 2017*; *Jin and Klionsky, 2014*). Although ATG9 has been shown to shuttle between the surface of mitochondria to the phagophore assembly site in yeast (*Reggiori et al., 2005*; *He and Klionsky, 2007*), the Golgi apparatus has been shown to be the major source of ATG9 pools, particularly in other eukaryotes (*Tung et al., 2010*; *Takahashi et al., 2011*; *Young et al., 2006*). In a previous study, TgATG9 was not found to co-localize with mitochondria in tachyzoites, but did co-localize with the Golgi and other organelles including the VAC, suggesting these as potential sources of TgATG9 in this parasite (*Nguyen et al., 2017*). Ablation of TgATG9 was not found to have an extensive impact on mitochondrial morphology in tachyzoites, but it was not assessed at the ultrastructural level (*Nguyen et al., 2017*). Here, our TEM findings of TgATG9-deficient bradyzoites frequently showed multiple fragmented and elongated mitochondria within a single parasite. Furthermore, we also found that morphological differences in mitochondria structure were only detectable in Δ*atg9* parasites by immunofluorescence microscopy after 6 days of differentiation, when parasites are predominantly residing within a cyst and express bradyzoite markers. By contrast, no distinguishable differences in mitochondria structure were detected within the first 3 days of differentiation, indicating that the onset of this phenotype in Δ*atg9* parasites is in fully differentiated bradyzoites soon after conversion, rather than during the process of differentiation. An increase in mitochondrial elongation and fragmentation was shown to occur in Drosophila deficient in the ATG2/ATG18/ATG9 complex (*Xu et al., 2019*). Most intriguingly the mitochondrial elongations we observed in TgATG9-deficient *T. gondii* bradyzoites seemed to form mitochondrial vacuoles strikingly similar to 'm-vacuoles' previously reported in the protist, *Dictyostelium* (*Matsuyama and Maeda, 1998*; *Tasaka and Maeda, 1983*), whereby the mitochondria has been shown to elongate with bulbous ends and adopts a horseshoe conformation that encapsulates the cytoplasm within differentiating prespore cells. Spherical mitochondrial structures have also been observed in mammalian cells treated with CCCP (carbonyl cyanide *m*-chlorophenyl hydrazone) (*Ding et al., 2012*; *Miyazono et al., 2018*), which collapses the mitochondrial membrane potential. These spherical structures were reported to encapsulate a variety of cytoplasmic structures and organelles, which is similar to the apparent sequestration of endoplasmic reticulum (ER), secretory organelles, lipid droplets, and even parts of the mitochondrion itself in Δ*atg9* bradyzoites. It is worth noting that previous reports have implicated ATG8 in organelle clearance in Plasmodium spp., which is essential to parasite development (*Voss et al., 2016*; *Jayabalasingham et al., 2010*). We speculate that the encapsulation of organelles by multimembrane vesicles (including apparent m-vacuoles) observed here could represent an attempt at organelle clearance in the absence of ATG9-dependent canonical autophagy in the related apicomplexan *T. gondii*. However, the extent to which the abnormal mitochondria are a consequence of or a response to deficient autophagy remains to be determined.

Expectedly, most conservation of *T. gondii* autophagy-related proteins was found in the closely related Sarcocystidia *H. hammondi*, *N. caninum*, and *Sarcocystis neurona*. While we identified matches across apicomplexans to many of the *T. gondii* autophagy-related proteins that are involved in maintenance of the apicoplast, little to no conservation was identified in parasites outside of the coccidia for apicoplast-independent canonical autophagy proteins (e.g., ATG1, ATG2, ATG9, and Prop1). The potential conservation of canonical autophagy in coccidia supports the hypothesis that this pathway promotes survival within the cysts that persist either in tissues or in the environment. Although we did not identify canonical autophagy genes in Cryptosporidia, which also use environmentally resilient oocysts for fecal-oral transmission, such parasites are known to have a severely reduced genome and thus might have evolved other strategies for long-term survival. Not only did we identify matches to *T. gondii* ATG1, ATG9, and TgProp1 in *C. velia* and *V. brassicaformis*, but we also found that many of the apicoplast-related autophagy proteins had better matches to proteins in these non-parasitic, photosynthetic protists than they did to other apicomplexan parasites. *C. velia* and *V. brassicaformis* were originally identified as endosymbionts of Scleractinia (stony corals) and

represent some of the closest known non-parasitic relatives of the Apicomplexa (*Oborník and Lukeš, 2015*; *Cumbo et al., 2013*; *Oborník et al., 2012*). Although it is not clear what role autophagy plays in their life cycle, matches for canonical autophagy proteins between these endosymbiotic relatives and certain apicomplexans indicate that canonical autophagy was conserved in a number of cyst-forming apicomplexans from a common ancestor and was lost in several apicomplexans that do not form tissue cysts (e.g., Cryptosporidium spp., Plasmodiumspp., and Babesia spp.).

Although our findings suggest a critical role for TgATG9 in canonical autophagy and persistence, virtually nothing is known about the initiation and biogenesis of autophagic structures in *T. gondii*. Gaining such insight will require the identification and interrogation of other autophagy-related proteins that function early in the pathway including the TgATG1 and TgATG2 homologs. Identifying other, potentially divergent, components of the early pathway will permit their evaluation as candidates for selectively disrupting the pathway to curb chronic infection.

# Materials and methods

## Ortholog identification
We sought to identify putative orthologs of *T. gondii* autophagy-related proteins across a number of closely related apicomplexan parasites (*H. hammondi, N. caninum, S. neurona,* Eimeriaspp., *Cyclospora cayetanensis, P. falciparum,* Babesiaspp., and *C. parvum*) and distantly related photosynthetic, non-pathogenic relatives (*C. velia* and *V. brassicaformis*). Amino acid sequences for *T. gondii* autophagy-related proteins were retrieved from ToxoDB.org and BLASTed against online databases for the relevant species listed above (ToxoDB.org, PlasmoDB.org, PiroplasmaDB.org, and CryptoDB.org). Sequence hits with E values of less than $10^{-3}$ were considered a putative match. Negative logarithms of the E values were plotted into a heat map to demonstrate conservation of autophagy-specific proteins in cyst-forming protozoans. To perform a negative logarithm transformation on highly conserved hits that returned E values of 0.0, these E values were set at $10^{-200}$.

## Cell lines
The host cell line used throughout this study was human foreskin fibroblasts (HFFs).

*T. gondii* cell lines used include the following: the unmodified Pru strain and the PruΔ*hxg* strain, both of which were modified as part of this study, as described below.

Uninfected host cells and host cells infected with the different strains are routinely tested for mycoplasma contamination using the Mycolasma Detection Kit from Southern Biotechnologies (13100–01).

## Generation of the S/ATG8 *T. gondii* strain
The *SAG1* promoter cassette was amplified using primers ML2475/ML2476 containing SpeI and XbaI restrictions sites, respectively, and cloned into the DHFR-TetO7 vector (Morlon-Guyot et al., Cell Microbiol. 2014) to yield the DHFR-pSAG1 plasmid. Then, using this plasmid as a template, the DHFR-pSAG1 cassette was amplified by PCR with primers ML2669/ML2670 and cloned downstream of the GFP-coding cassette in the NsiI-digested pGFP-TgAtg8 plasmid (*Besteiro et al., 2011*). Clones were verified for correct insert orientation by sequencing. Using this plasmid as a template, a 3706 bp fragment containing the DHFR expression cassette, the SAG1 promoter, and the GFP-coding fragment was amplified by PCR using the KOD polymerase (Merck) and the ML2477/ML2664 primers that included regions for integration by double homologous recombination. A protospacer sequence targeting the native TgATG8 promoter was generated by annealing primers ML2885/ML2886 and inserting the corresponding fragment at the BsaI site of vector pU6-Cas9 (*Sidik et al., 2016*). This plasmid was co-transfected with the DHFR-pSAG1-GFP donor sequence in tachyzoites of the Pru strain for integration by CRISPR/CAS9, and transgenic parasites were then selected with pyrimethamine and cloned by limiting dilution.

## Generation of the TgATG9 knockout cell line
The HXGPRT cassette was amplified from the pmini-HXGPRT plasmid (*Donald and Roos, 1998*) using primers ML2465/ML2466 containing the HindIII and BamHI restriction sites, respectively. This cassette was then inserted into the HindIII-BamHI-digested pTub5/CAT-TgAtg9 plasmid, which was

previously generated for creating a TgATG9 knockout in an RH strain and thus already contained 5′ and 3′ TgATG9 homology regions for gene replacement (Nguyen et al., Cell Microbiol. 2016). The resulting HXGPRT-TgATG9 plasmid was then linearized with KpnI/NotI. A protospacer sequence targeting the 5′ of the TgATG9 coding sequence was generated by annealing primers ML2467/ML2468 and inserting the corresponding fragment at the BsaI site of vector pU6-Cas9 (*Sidik et al., 2016*). The linearized HXGPRT-TgATG9 plasmid, serving as a donor sequence, and the TgATG9-targeting Cas9-expressing plasmid were co-transfected in ΔHXGPRT Pru tachyzoites. Transgenic parasites were then selected with mycophenolic acid and xanthine, and subsequently cloned by limit dilution.

## Genetic complementation

The coding sequence of the *TgATG9* gene was amplified from a *T. gondii* cDNA library (derived from the ME49 strain) using primers DS6/DS7. The plasmid from *Di Cristina et al., 2017* was amplified using primers D5/D8. The fragments generated from these PCRs had compatible ends, allowing insertion of the *ATG9* fragment into the plasmid by Gibson assembly, in place of where *TgCPL* is in the template plasmid. A 1500 bp sequence of the *TgATG9* promoter was amplified from genomic DNA of the Pru*Δhxg* strain using primers DS17/18 and ligated into the *ATG9* plasmid immediately upstream of the 5′ end of *ATG9*. A 3×HA tag was inserted onto the 3′ end of *ATG9* by PCR using the primers DS24/DS25. The plasmid was then linearized by digestion with PciI and used to transfect $1 \times 10^7$ *Δatg9* tachyzoites. The linearized plasmid was incorporated into the genome of *Δatg9* tachyzoites by random integration. Based on the presence of a resistance cassette in the transfection plasmid, positively transfected parasites were selected using bleomycin prior to isolating clones by limiting dilution. Clonal populations were screened for TgATG9 expression by probing fixed parasites with α-HA antibodies.

## Bradyzoite differentiation

For all bradyzoite conversion, tachyzoites were mechanically lysed by scraping infected HFF monolayers that were then passed sequentially through 20G and 23G syringes and a 3 μm filter. Filtered parasites were then counted and allowed to infect fresh monolayers of HFF cells for 24 hr. Bradyzoite differentiation was induced using alkaline pH medium and ambient $CO_2$ (*Soete et al., 1993*; *Soête et al., 1994*; *Weiss et al., 1995*). Briefly, 24 hr after parasites were applied to HFF monolayers, DMEM media was replaced for an alkaline differentiation media (RPMI without $NaHCO_3$, 50 mM HEPES, pen/strep, 1% FBS, pH 8.25). Differentiation media was replaced daily.

## Immunoblotting

Tachyzoites were harvested while they were still largely intracellular. Bradyzoites were generated by induced differentiation with 1% FBS for 7 days in ambient $CO_2$ condition. Parasites were lysed in RIPA buffer for 10 min at 4°C with shaking and pelleted by centrifuging at 10,000 *g* for 10 min at 4°C. Supernatants were collected and mixed with sample buffer and β-mercaptoethanol. Protein lysates from ~2.5 × $10^6$ parasites were subjected to electrophoresis on 7.5% SDS-PAGE gels and transferred onto 0.45 μm nitrocellulose membranes (Bio-Rad, cat. # 1620115). Buffer containing a high concentration of both Tris and glycine (50 mM Tris, 380 mM glycine, 0.1% SDS, and 20% methanol) was used for transferring proteins to the membrane over the course of 10 hr at 20 V at 4°C. Following transfer, the membrane was blocked with 5% milk in PBS (with 0.05% Tween X-114) for 30 min at room temperature (RT). Primary antibodies were diluted in wash buffer (1% milk in PBS with 0.05% Tween 20) and applied to membranes overnight at 4°C. The primary antibodies used include mouse anti-HA (1:2500; clone 16B12, Biolegend, cat. # 901501), mouse anti-GFP (1:500; clones 7.1 and 13.1, Roche, cat. # 11814460001), rabbit anti-TgATG8 (1:500; *Besteiro et al., 2011*), mouse anti-α-tubulin (1:2,000; clone B-5-1-2, Sigma-Aldrich, cat. # T5168), and mouse anti-MIC2 (1:2500; 6D10). Membranes were washed three times with wash buffer before incubation with HRP-conjugated secondary antibodies (1:2500) for 1 hr at RT. Proteins were detected using SuperSignal West Pico PLUS Chemiluminescent Substrate (Thermo Fisher Scientific, cat. #1863096) or SuperSignal West Femto (Thermo Fisher Scientific, cat. #34094). The Syngene Pxi imaging system was used to detect signals.

## Staining autophagic material with CytoID

Tachyzoites were differentiated to bradyzoites for 7 days and then treated with either 1 µM LHVS or an equal volume of DMSO for 1 day. The CytoID Autophagy Detection Kit 2.0 (Enzo) was used to stain autophagosomes within live bradyzoites prior to fixation with 4% paraformaldehyde, following the manufacturer's instructions, and the cyst wall was stained using Rhodamine labeled *Dolichos biflorus* agglutinin (1:400; Vector Laboratories, cat. # RL-1032). The number and size of CytoID-positive structures were measured using CellProfiler (*Carpenter et al., 2006*). The boundaries of each cyst were identified manually based on Dolichos lectin staining. Raw images for CytoID staining were first corrected for uneven illumination/lighting/shading to reduce uneven background signal. CytoID signal was identified using Otsu two-class thresholding method. The image processing pipeline is available at: https://cellprofiler.org/examples/published_pipelines. Measurements of CytoID puncta were done automatically within CellProfiler. The definition of all the measurements can be found at: http://cellprofiler-manual.s3.amazonaws.com/CellProfiler-3.0.0/modules/measurement. html.

For probing autophagic material in S/Atg8 bradyzoites, tachyzoites were converted for 7 days. After this time LHVS (1 µM) was added for either 1 or 7 days. An equal volume of DMSO was added as a vehicle control treatment. CytoID was then performed as described above.

## Immunofluorescence

For IFAs, HFF monolayers were grown overnight on cover slips, then infected with parasites for 1 hr before they were washed twice with 37°C PBS, fixed for 20 min with 4% (w/v) paraformaldehyde in PBS, permeabilized with 0.1% Triton X-100 in PBS for 15 min, and blocked with 0.1% (w/v) fatty acid-free BSA (Sigma-Aldrich, cat. # 9048-46-8) in PBS. Rat anti-HA (Sigma-Aldrich, cat. # 11867423001) (1:500), mouse anti-GFP (1:500; clones 7.1 and 13.1, Roche, cat. # 11814460001), mouse MAb 45.56 anti-TgIMC1 (1:500; Gary Ward, University of Vermont), with mouse MAb anti-$F_1F_0$ATPase (1:2000; P. Bradley, UCLA), and rabbit anti-TgCPN60 (1:2000; *Agrawal et al., 2009*) were used for 1 hr of primary antibody staining in wash buffer consisting of 1% cosmic calf serum in PBS. Cover slips were rinsed three times and then washed three times for 5 min in wash buffer. Secondary antibodies were goat anti-rat Alexa Fluor 594 (Invitrogen, cat. # A11007) (1:1000), goat anti-rabbit Alexa Fluor 594 (Invitrogen, cat. # A11012), goat anti-rabbit Alexa Fluor 488 (Invitrogen, cat. # A11008), and goat anti-streptavidin Alexa Fluor 350 (1:1000). They were used for 1 hr diluted in wash buffer at 1:1000. DAPI (Sigma-Aldrich, cat. # D9542) was used at 1:200. Fluorescein and biotinylated *D. biflorus* agglutinin were used at 1:400 (Vector Laboratories, cat. # FL-1031 and B-1035). Cover slips were then washed three times for 5 min in wash buffer and subsequently mounted on slides with 8 µl of prolong glass (Invitrogen, cat. # P36984) or mowiol. Images were taken on a Zeiss Axio Observer Z1 inverted microscope at 100× and analyzed using Zen 3.0 blue edition software.

## Bradyzoite puncta

The accumulation of dense material within in vitro bradyzoite cysts was quantitatively measured using the puncta quantification assay, as described previously (*Di Cristina et al., 2017*). Briefly, HFF cell monolayers were grown on cover slips and infected with *T. gondii* tachyzoites. Twenty-four hours post-invasion, intracellular tachyzoites were differentiated to bradyzoites over the course of 7 days (as described above) and subsequently treated with 1 µM LHVS or 0.1% DMSO (vehicle control) for 24 hr. For direct immunofluorescence of bradyzoite cyst walls, *D. biflorus* agglutinin conjugated with fluorescein or rhodamine (Vector Laboratories, cat. #s FL-1031 and RL-1032–2, respectively) was diluted 1:400 and incubated with infected monolayers that had been fixed with 4% paraformaldehyde and made permeable with Triton X-100. The fluorescent signal derived from DBA staining allows automatic detection of the bradyzoite-containing cyst area using ImageJ software. Cyst images were captured at 100× or 63× oil objective (three biological replicates at each objective) as described above. Automatic bradyzoite cyst identification and puncta quantification were performed in ImageJ, using the following parameters described previously by *Di Cristina et al., 2017*. Maximum entropy thresholding on the GFP channel was used to identify cysts. This was followed by the identification of objects with areas between 130 and 1900 µm². Particles (puncta) under the GFP mask (therefore within a cyst) were analyzed by automatic local thresholding on the phase image using the Phansalkar method, with the following parameters: radius = 5.00 µm; k value = 0.5; r

value = 0.5. Puncta were measured from the resulting binary mask by particle analysis according to the following: size = 0.3–5.00 µm; circularity = 0.50–1.00.

## Bradyzoite differentiation assay

HFF monolayers were grown on cover slips in six-well tissue culture plates and subsequently infected with tachyzoites for 24 hr. Invaded parasites were differentiated to bradyzoites as described above for up to 4 days. Samples were washed with PBS and fixed with 4% (w/v) paraformaldehyde after 1, 2, 3, or 4 days of differentiation. Fixation, permeabilization, and antibody staining were performed as described above. To probe for tachyzoite and bradyzoite-specific markers, samples were probed with rat anti-SAG1 (1:1000) and rabbit anti-BAG1 (1:1000). Secondary antibodies used were goat anti-rat Alexa Fluor 594 (1:1000) and goat anti-rabbit Alexa Fluor 488 (1:1000). Cover slips were then washed and incubated with mouse anti-GRA7 (1:2000) followed by goat anti-mouse Alexa Fluor 350 (1:1000). Images were taken on a Zeiss Axio Observer Z1 inverted microscope at 63× and analyzed using Zen 3.0 blue edition software. The exported tiff files were run through a CellProfiler pipeline that identified PV/cysts based on GRA7 signal through the IdentifyPrimaryObjects module and the object size was measured with the MeasureObjectIntensity module. The intensity of the SAG1 (red channel) and BAG1 (green channel) staining for each object was measured using the MeasureObjectIntensity module. Using the median fluorescence intensity, the ratio of the two signals was normalized in R, and objects that were below 2000 pixels in size or above an eccentricity of 0.9 were removed. The ratiometric SAG1:BAG1 data was $\log_2$ transformed and graphed using Prism v8.0.

## Quantification of mitochondrial morphology

Tachyzoites were seeded onto HFF monolayers in a 96-well plate and differentiated for up to 8 days, as described above. Samples were then fixed, permeabilized, and stained with rabbit anti-IMC1, mouse MAb anti-$F_1F_0$ATPase, and Dolichos lectin, as described above. Ten representative fields of view per condition were captured with the Zeiss Axio Observer Z1 using a 100× objective. Images were subsequently analyzed in CellProfiler to detect cysts (demarcated using IMC staining) and the granularity and texture modules in CellProfiler were used to capture structural elements and intensity characteristics of the $F_1F_0$ATPase staining, respectively. Next, $F_1F_0$ATPase measurements were analyzed in KNIME. Random forest analysis was used to determine the morphological measurements that distinguish $\Delta atg9$ from Pru$\Delta hxg$ and $\Delta atg9ATG9$ on day 8 (listed in *Supplementary file 2*). Principal component analysis was then used to reduce the dimensionality of the measurement data ($F_1F_0$ATPase Morphological Score).

## TEM of in vitro bradyzoite cysts

Sample preparation for TEM was carried out as described by *Di Cristina et al., 2017*. Briefly, HFF monolayers grown in six-well plates were infected with $5 \times 10^4$ tachyzoites in D10 media. After O/N incubation at 37˚C, 5% $CO_2$, the D10 medium was replaced with alkaline media (pH 8.2) to induce bradyzoite conversion. Alkaline media was changed daily over the course of 7 days, replacing with fresh alkaline media and plates were incubated at 37˚C with 0% $CO_2$. TEM preparation of samples was carried out by washing the infected monolayers with cold PBS three times, followed by fixation with 2.5% glutaraldehyde (EMS, cat. # 16210) in 0.1 M sodium cacodylate buffer (pH 7.4) for 1 hr at RT. Fixed infected monolayers then were gently lifted using a cell scraper to detach large sheets and transferred to microcentrifuge tubes. Samples were centrifuged at 1500 *g* for 10 min at RT and washed three times with 0.1 M sodium cacodylate buffer (pH 7.4). Samples were stored in the same buffer at 4˚C until processed for TEM as described in *Coppens and Joiner, 2003* before examination with a Philips CM120 electron microscope under 80 kV.

## Bradyzoite viability

Bradyzoite viability was assessed by combining plaque assay and qPCR analysis of genome number, as previously described by *Di Cristina et al., 2017*. Briefly, HFF cell monolayers were infected with *T. gondii* tachyzoites in six-well plates. Following host cell invasion, tachyzoites underwent differentiation to bradyzoites as described above, resulting in the generation of in vitro tissue cysts. Differentiation was carried out over the course of 7 days, replacing the alkaline media daily. Following differentiation, parasites were treated with 1 µM LHVS or 0.1% DMSO (vehicle control) in

differentiation media. Treatment was replaced daily for 7 and 14 days, respectively. Following the treatment period, the culture media in each well was replaced with 2 ml Hanks Balanced Salt Solution and cysts were liberated from the infected HFF monolayers by mechanical extrusion, by lifting cells with a cell scraper and syringing several times through 25G needles. Then, 2 ml of pre-warmed 2 × pepsin solution (0.026% pepsin in 170 mM NaCl and 60 mM HCl, final concentration) was added to each sample and samples left to incubate at 37°C for 30 min. Reactions were stopped by adding 94 mM $Na_2CO_3$, removing the supernatant after centrifugation at 1500 $g$ for 10 min at RT, and resuspending pepsin-treated parasites in 1 ml of DMEM without serum. Parasites were enumerated and 1500 parasites per well were added to six-well plates containing confluent monolayers of HFFs in D10 media, in triplicate. To allow for the formation of bradyzoite-derived plaques, parasites were left to grow undisturbed for 12 days. After this period, the number of plaques in each well was determined by counting plaques with the use of a light microscope. Five hundred μl of the initial 1 ml of pepsin-treated parasites was used for genomic DNA purification, performed using the DNeasy Blood and Tissue Kit (Qiagen). Genomic DNA was eluted in a final volume of 200 μl. To determine the number of parasite genomes per microliter, 10 μl of each gDNA sample was analyzed by qPCR, in duplicate, using the tubulin primers TUB2.RT.F and TUB2.RT.R (*Kannan et al., 2019*). qPCR was performed using Brilliant II SYBR Green QPCR Master Mix (Agilent) and a Stratagene Mx3000PQ-PCR machine. The number of plaques that formed was then normalized to the calculated number of genomes present in the inoculating sample. Despite this normalization, these experiments can have considerable inter-experiment variation, which in this case necessitated secondary normalization to PruΔ*hxg*-DMSO within each biological replicate. Because secondary normalization to PruΔ*hxg*-DMSO precluded statistical comparison of this group to the others, we repeated the experiment and assessed viability after 14 days of differentiation without treatment.

## Mouse infection experiments

Seven-week-old female CBA/J mice (Jackson) were randomly assigned to groups and infected intraperitoneally (i.p.) with 150 tachyzoites of PruΔ*hxg*, Δ*atg9*, or Δ*atg9ATG9* strains. Group sizes in each experiment are as follows: In the first experiment, 10 mice were infected with one of the three strains, five mice per strain were humanely euthanized 1 week post-infection to assess brain tachyzoite burden, five mice per strain were humanely euthanized 5 weeks post-infection to assess brain cyst burden. The second experiment was set up the same as the first experiment, however an additional 10 mice were also infected with PruΔ*hxg* and their survival assessed. Assessing the statistical differences in parasite burden between mice infected with each strain was complicated by the absence of data from the mice that did not survive infection. Because mice that die from the infection often have higher burden and become moribund and must be euthanized before the 5-week time point when cysts are counted, enumeration of cysts from those that survive probably underestimates differences. Data were pooled from the two independent experiments. The mouse brains were placed in a set volume of sterile PBS that provided a concentration of 500 mg of mouse brain per ml. Mouse brains were individually minced with scissors, vortexed and homogenized by three or four passages through a 21G syringe needle. Three 10 μl samples (30 μl total) of brain homogenates per infected mouse were analyzed by phase-contrast microscopy to enumerate cysts and the number of cysts per brain determined (scaled appropriately to the total volume of brain homogenate for that mouse). Samples for which no cysts were observed in the 30 μl of homogenate were given a value (17 cysts/brain) that is half the limit of detection (<33 cysts/brain). Mouse sample sizes were chosen based on previous studies. Mice were randomly assigned to groups and samples were blinded for enumeration of cysts. Animal studies described here adhere to a protocol approved by the Committee on the Use and Care of Animals of the University of Michigan.

Genomic DNA was extracted from 25 mg of brain homogenate tissue using the DNeasy Blood and Tissue Extraction kit (Qiagen) following the manufacturer's instructions. SYBR Green qPCR was performed on gDNA using 300 nM Tox9 and Tox11 primers and reaction conditions previously described (*Kannan et al., 2019*).

## Tachyzoite plaque assays

Intracellular tachyzoites were purified from HFFs following standard protocols. Parasites were added to and left undisturbed on a monolayer of HFFs for 10 days, after which time plaques were counted.

An aliquot of purified parasites was collected for gDNA extraction to normalize tachyzoite plaque numbers. Briefly, gDNA was extracted using the Phire Tissue Direct PCR Master Kit (Thermo Fisher Scientific; *Piro et al., 2020*). SYBR Green qPCR was performed for $\alpha$-tubulin following conditions previously used (*Kannan et al., 2019*).

### Statistical analysis

Data were analyzed after removing outliers using ROUT with a Q value of 0.1% and testing for normality and equal variance. Since all data failed either test, a Kruskal-Wallis test with Dunn's multiple comparisons was used to compare groups within the same genotype or treatment. Mouse survival was analyzed using a Mantel-Cox log-rank test.

Power analysis was not performed when the study was designed. Sample sizes were determined based on the capacity of each particular assay to collect a sufficient number of values to support rigorous analyses using standard statistical tests.

## Acknowledgements

We thank Christian McDonald and all members of the Carruthers lab for providing feedback on the study. The authors appreciate My-Hang (Mae) Huynh's help with proofreading the manuscript. We also thank Mack Reynolds for help with CellProfiler, Aric Schultz for help with ImageJ, Michael Delannoy and other staff at the Johns Hopkins University Microscopy facility, and Drs Peter Bradley, Boris Striepen, and Gary Ward for generously providing antibodies for this study. We also appreciate the guidance provided by Drs Jonny Sexton, Matthew O'Meara, and Daniel Klionsky. This work was supported by National Institutes of Health grants R01AI120627 (VBC and MDC) and R01AI060767 (IC) and a post-baccalaureate fellowship R25GM086262 (NMS). The study was also conducted with support from the Agence Nationale de la Recherche (ANR-19-CE15-0023), the Fondation pour la Recherche Médicale (FRM EQ20170336725) (SB), and the University of Perugia Fondo Ricerca Di Base 2019 program of the Department of Chemistry, Biology and Biotechnology (MDC).

## Additional information

### Funding

| Funder | Grant reference number | Author |
| --- | --- | --- |
| National Institutes of Health | R01AI120627 | Vern B Carruthers |
| National Institutes of Health | R01AI060767 | Isabelle Coppens |
| Agence Nationale de la Recherche | ANR-19-CE15-0023 | Sébastien Besteiro |
| Fondation pour la Recherche Médicale | FRM EQ20170336725 | Sébastien Besteiro |
| National Institutes of Health | R25GM086262 | Nayanna M Mercado Soto |

The funders had no role in study design, data collection and interpretation, or the decision to submit the work for publication.

### Author contributions

David Smith, Data curation, Formal analysis, Investigation, Methodology, Writing - original draft; Geetha Kannan, Fengrong Wang, Einar B Olafsson, Data curation, Formal analysis, Investigation, Methodology; Isabelle Coppens, Data curation, Formal analysis, Funding acquisition, Investigation, Methodology; Hoa Mai Nguyen, Aude Cerutti, Data curation, Formal analysis, Investigation; Patrick A Rimple, Data curation, Investigation; Tracey L Schultz, Investigation; Nayanna M Mercado Soto, Data curation; Manlio Di Cristina, Resources, Data curation, Formal analysis, Investigation, Methodology; Sébastien Besteiro, Conceptualization, Formal analysis, Funding acquisition; Vern B Carruthers, Conceptualization, Supervision, Funding acquisition, Writing - original draft, Project administration

## Author ORCIDs
David Smith https://orcid.org/0000-0002-5158-0522
Manlio Di Cristina http://orcid.org/0000-0003-4154-5210
Sébastien Besteiro http://orcid.org/0000-0003-1853-1494

## Ethics
Animal experimentation: All laboratory animal work in this study was carried out in accordance with policies and guidelines specified by the Office of Laboratory Animal Welfare, the US Department of Agriculture, and the American Association for Accreditation of Laboratory Animal Care (AAALAC). The University of Michigan Committee on the Use and Care of Animals (IACUC) approved the animal protocol used for this study (Animal Welfare Assurance A3114-01, protocol PRO00008638).

## Decision letter and Author response
Decision letter https://doi.org/10.7554/eLife.59384.sa1
Author response https://doi.org/10.7554/eLife.59384.sa2

# Additional files
## Supplementary files
- Supplementary file 1. Primers used in this study.
- Supplementary file 2. CellProfiler parameters for determining mitochondria morphology.
- Transparent reporting form

## Data availability
All data generated or analysed during this study are included in the manuscript and supporting files.

The following previously published datasets were used:

| Author(s) | Year | Dataset title | Dataset URL | Database and Identifier |
|---|---|---|---|---|
| Gajria B, Bahl A, Brestelli J, Dommer J, Fischer S, Gao X, Heiges M, Iodice J, Kissinger JC, Mackey AJ, Pinney DF, Roos DS, Stoeckert CJ, Wang H, Brunk PB | 2007 | ToxoDB | https://toxodb.org/toxo/ | toxodb, 10.1093/nar/gkm981 |
| Aurrecoechea C, Brestelli J, Brunk BP, Dommer J, Fischer S, Gajria B, Gao X, Gingle A, Grant G, Harb OS, Heiges M, Innamorato F, Iodice J, Kissinger JC, Kraemer E, Li W, Miller JA, Nayak V, Pennington C, Pinney DF, Roos DS, Ross C, Stoeckert CJ, Treatman C, Wang H | 2008 | PlasmoDB | https://plasmodb.org/plasmo/ | plasmodb, 10.1093/nar/gkn814 |
| Heiges M, Wang H, Robinson E, Aurrecoechea C, Gao X, Kaluskar N, Rhodes P, Wang S, | 2006 | CryptoDB | https://cryptodb.org/cryptodb/ | cryptodb, 10.1093/nar/gkj078 |

| He CZ, Su Y, Miller J, Kraemer E, Kissinger JC | | | | |
|---|---|---|---|---|
| Aurrecoechea C, Brestelli J, Brunk BP, Fischer S, Gajria B, Gao X, Gingle A, Grant G, Harb OS, Heiges M, Innamorato F, Iodice J, Kissinger JC, Kraemer ET, Li W, Miller JA, Nayak V, Pennington C, Pinney DF, Roos DS, Ross C, Srinivasamoorthy G, Stoeckert CJ, Thibodeau R, Treatman C, Wang H | 2010 | PiroplasmaDB | https://piroplasmadb.org/piro/ | piroplasmadb, 10.1093/nar/gkp941 |

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
