## [Decision Letter]

**Acceptance summary:**

Apicomplexan parasites have co-opted elements of the autophagy pathway to fulfil phylum-specific functions. While previous studies had implicated certain autophagy-related genes as necessary for the correct segregation of the apicoplast, the role of other canonical autophagy pathways has remained largely uncharacterized. Your study now presents conclusive evidence showing that autophagy is necessary for the survival of chronic stages of *Toxoplasma gondii*. The results provide shed light on critical aspects of parasite biology and the fundamental role of autophagy therein.

**Decision letter after peer review:**

Thank you for submitting your article "Toxoplasma TgATG9 is critical for autophagy and long-term persistence in tissue cysts" for consideration by *eLife*. Your article has been reviewed by 3 peer reviewers, one of whom is a member of our Board of Reviewing Editors, and the evaluation has been overseen by Dominique Soldati-Favre as the Senior Editor. The reviewers have opted to remain anonymous.

The reviewers have discussed the reviews with one another and the Reviewing Editor has drafted this decision to help you prepare a revised submission.

Summary:

Apicomplexan parasites have co-opted elements of the autophagy pathway to fulfil phylum-specific functions. While previous studies had implicated certain autophagy-related genes as necessary for the correct segregation of the apicoplast, the role of other canonical autophagy pathways has remained largely uncharacterized. Previous indirect evidence from parasites lacking one of the major proteases (TgCPL) in their endolysosomal compartment (VAC) suggested the occurrence of autophagy during the chronic stages (bradyzoites) of their asexual cycle. Such chronic stages are refractory to currently available drugs and clearance by the immune system, making potentially essential pathways of broad clinical interest. The authors show here that expression of ATG8 under the control of an acute-stage promoter leads to the accumulation of autophagic material in the bradyzoite VAC and fewer viable bradyzoites following in vitro differentiation. They also created and phenotypically characterized a parasite line lacking TgATG9, which they argue is exclusively involved in canonical autophagy-in contrast to TgATG8, which also plays a role in apicoplast segregation. Through these and other in vitro and in vivo studies, the authors conclude that (i) canonical autophagy mediated by ATG9 occurs in bradyzoites, and (ii) that this pathway is required for long-term persistence in tissue cysts. The evidence for the first claim is conclusive; however, the second claim needs to be supported by evidence that distinguishes between defects in bradyzoite differentiation and the titular claim of bradyzoite persistence. Although addressing this major point would be sufficient for publication, examining the mechanistic basis for the requirement for canonical autophagy in bradyzoites (e.g. amino acid recycling, organelle quality control, other cellular processes) would increase the overall impact of the study and clarify the role of autophagy in bradyzoite persistence.

Essential revisions:

1. Experimentally distinguish between defects in differentiation and persistence in the ATG9 KO. This could be accomplished by quantitatively demonstrating that differentiation proceeds normally (with similar rates and kinetics) in vitro. Alternatively, showing that brain cysts form at similar rates, but are lost over time in the ATG9 KO, would similarly address the concern.

2. As suggested above, mechanistically understanding why bradyzoites require autophagy, while tachyzoites do not, could provide a mechanistic link between autophagy and persistence (as distinct from differentiation). What are the cellular contents that must be recycled by autophagy to support bradyzoite development and viability? Experiments aimed at differentiating amino acid usage, mitochondria and/or ER recycling by autophagy to tease out these different requirements have not been performed and/or discussed. Along similar lines, the temporal relationship between the onset of differentiation and the various phenotypes (mitochondrial fragmentation, cellular defects) could be better established, and the authors could use available markers to determine whether the residual CytoID staining in the ATG9 KO parasites colocalizes with ATG8 or the VAC, and how that pattern compares to the wild-type or complemented parasites.

3. Statistics. Please consider whether the statistics are appropriate when comparing cells vs. independent experiments. Does the distribution of values in each dataset conform with these requirements of the nonparametric tests used?

---

## [Author Response]

Essential revisions:1. Experimentally distinguish between defects in differentiation and persistence in the ATG9 KO. This could be accomplished by quantitatively demonstrating that differentiation proceeds normally (with similar rates and kinetics) in vitro. Alternatively, showing that brain cysts form at similar rates, but are lost over time in the ATG9 KO, would similarly address the concern.

In response to the valid point raised by the reviewer, we performed in vitro bradyzoite differentiation assays with WT, ∆*atg9* and ∆*atg9ATG9* strains. Each strain was evaluated for expression of SAG1 (tachyzoite specific) and BAG1 (bradyzoite specific) by IFA 1-4 days after the start of differentiation. Images were taken of multiple PVs/cysts (at least 134 per sample type across 3 biological replicates) and the SAG1 and BAG1 intensity was measured using CellProfiler. We reported a ratio of SAG1:BAG1 as our readout for parasite differentiation. The ratiometric data was subject to a log_2_ transformation and subsequently plotted. Higher positive values indicate more SAG1 relative to BAG1 signal and conversely, higher negative values indicate more BAG1 relative to SAG1 signal. This data is presented in the new Figure 2—figure supplement 1. Overall, we found that ∆*atg9* parasites do not have a defect in differentiation. This suggests that ATG9-dependent autophagy is not necessary for differentiation, but rather is needed for parasite viability after stage conversion.

2. As suggested above, mechanistically understanding why bradyzoites require autophagy, while tachyzoites do not, could provide a mechanistic link between autophagy and persistence (as distinct from differentiation). What are the cellular contents that must be recycled by autophagy to support bradyzoite development and viability? Experiments aimed at differentiating amino acid usage, mitochondria and/or ER recycling by autophagy to tease out these different requirements have not been performed and/or discussed. Along similar lines, the temporal relationship between the onset of differentiation and the various phenotypes (mitochondrial fragmentation, cellular defects) could be better established, and the authors could use available markers to determine whether the residual CytoID staining in the ATG9 KO parasites colocalizes with ATG8 or the VAC, and how that pattern compares to the wild-type or complemented parasites.

Determining the cellular contents for recycling via autophagy in bradyzoites (i.e., determining which substrates are used) will require the identification of specific autophagy cargo adaptor proteins. Since such proteins are not evolutionarily conserved and it will likely take years to identify them, we feel this is well beyond the scope of the current study. Similarly, although measurements of amino acids or other metabolites are becoming more feasible (albeit still technically challenging), such analysis is complicated by the non-synchronous defects seen in ∆atg9 bradyzoites, thus complicating the interpretation of the analysis. We anticipate the need to generate conditional knockouts (e.g., AID) that will allow rapid and synchronous depletion of ATG9 to permit distinguishing direct from indirect consequences of acutely impairing autophagy. The work herein focused on measuring the extent to which Atg9, and by extension autophagy, is required for bradyzoite persistence in vitro and in vivo and reports various structural defects associated with bradyzoite autophagy deficiency. With the inclusion of the differentiation assays that demonstrate Atg9 deficiency does not affect tachyzoite to bradyzoite differentiation, we have demonstrated that this process is particularly important for the persistence of bradyzoites rather than differentiation to this life stage. We feel this is an important conceptual advance, since it suggests that the parasite is susceptible to disruption of a homeostatic processes during persistence.

In response to the suggestion to investigate onset of the observed phenotypes, we were able to successfully interrogate this for mitochondria morphology. We developed an algorithm for use in CellProfiler that measured various imaging parameters of mitochondria (labelled with anti-F_1_F_0_ATPase) in parasites 1, 3, 6 and 8 days after exposure to conditions that stimulate tachyzoite to bradyzoite differentiation in vitro. The dimensionality of the different mitochondria measurements was reduced by principle component analysis and is now presented in Figure 6F,G. Here, we found that the abnormal mitochondria morphology was detectable after 6- and 8-days post-differentiation, but not after 1 or 3 days. Given that parasites become fully differentiated after ~5 days culture under differentiation culture conditions, this suggests that the onset of abnormal mitochondria morphology occurs in fully differentiated parasites and not in differentiating parasites. Together with our new data that shows Atg9-deficient parasites differentiate to bradyzoites at a normal rate, this indicates that Atg9 is important for bradyzoite fitness at a more chronic phase, as opposed to early in the differentiation process. This justifies and necessitates forthcoming studies that will probe mechanisms of autophagy in bradyzoites in greater detail. We made several attempts to quantify the temporal onset of the parasite cell division phenotype using widefield and confocal microscopy. However, because bradyzoites are so tightly packed together within cysts in 3 dimensions, the imaging software is incapable of accurately segmenting individual parasites. As such it was not possible to agnostically quantify the cell division defect. Nevertheless, the additional experiments confirmed that such a defect is readily apparent, particular after 6 days of differentiation.

We also made several attempts to determine the extent to which residual CytoID staining in ∆atg9 bradyzoites corresponded to CPL in the VAC or ATG8-positive autophagosomes. Unfortunately, the residual staining is so weak that we were unable to reliably determine its spatial relationship with markers for the VAC or autophagosomes. In the absence of having a mutant that has been validated to have no autophagy, it is not possible to determine if the residual staining is specific for autophagosomes or autolysosomes. We regret that this is a limitation of the currently available means of assessing autophagy in *T. gondii*.

3. Statistics. Please consider whether the statistics are appropriate when comparing cells vs. independent experiments. Does the distribution of values in each dataset conform with these requirements of the nonparametric tests used?

To help ensure that the statistical analyses were performed correctly, we consulted a PhD statistician at the Center for Statistics, Computing, and Analytics Research (CSCAR) at the University of Michigan. The consultation reaffirmed that appropriate statistical tests were applied to assess potential differences in measurements made throughout the study. As described in the methods section on statistical analyses, we applied non-parametric tests when the distributions of data did not conform to a normal distribution or had unequal variance. We believe that taking a conservative approach to the statistical analysis helps to ensure confidence in the conclusions that were rendered in the study.